# HIPPOFORMER: INTEGRATING HIPPOCAMPUS-INSPIRED SPATIAL MEMORY WITH TRANSFORMERS

**Tiantian Li**[2,*]    **Xingxing Cao**[1,*]    **Yifei Wang**[2]    **Xiaojiao Yang**[1]

**Xiaolong Zou**[1,✉]    **Bo Hong**[1,✉]

[1] Department of Biomedical Engineering, School of Medicine, Tsinghua University
[2] Qiyuan Lab
[*] Equal contribution, [✉] Corresponding author

## ABSTRACT

Transformers form the foundation of modern generative AI, yet their key–value memory lacks inherent spatial priors, constraining their capacity for spatial reasoning. In contrast, neuroscience points to the hippocampal–entorhinal system, where the medial entorhinal cortex provides structural codes and the hippocampus binds them with sensory codes to enable flexible spatial inference. However, existing hippocampus models such as the Tolman-Eichenbaum Machine (TEM) suffer from inefficiencies due to outer-product operations or context-length bottlenecks in self-attention, limiting their scalability and integration into modern deep learning frameworks. To bridge this gap, we propose mm-TEM, an efficient and scalable structural spatial memory model that leverages meta-MLP relational memory to improve training efficiency, form grid-like representations, and reveal an intriguing link between prediction horizon and grid scales. Extensive evaluation shows its good generalization on long sequences, large-scale environments, and multi-step prediction, with analyses confirming that its advantages stem from explicit understanding of spatial structures. Building on this, we introduce Hippoformer, which integrates mm-TEM with Transformer to combine structural spatial memory with precise working memory, achieving superior generalization in both 2D and 3D prediction tasks and highlighting the potential of hippocampal-inspired architectures for complex domains. Overall, Hippoformer represents a initial step toward seamlessly embedding structured spatial memory into foundation architectures, offering a potential scalable path to endow deep learning models with spatial intelligence.

## 1 INTRODUCTION

The Transformer architecture has driven the recent advances in generative AI, with systems such as ChatGPT as prominent examples. This success has made the search for new architectural designs a central direction in machine learning. Transformer can be viewed as associative memories implemented through key-value caches and self-attention retrieval(Vaswani et al., 2017; Geva et al., 2021; Ramsauer et al., 2020). However, they face inherent limitations, most notably quadratic computational cost and redundant memory, which limit their scalability(Zhuang et al., 2023). To address these issues, many alternatives have been proposed by reconsidering memory design. For instance, Titans leverage fast MLP weights for large-capacity(Behrouz et al., 2024). Although these approaches improve long-sequence modeling, their memory structures remain largely flat and lack a critical element: an inherent spatial memory. Such a memory, however, is vital for organizing the "what-where" of experiences and for building internal models Yang et al. (2025). Therefore, developing architectures with structured spatial memory and integrating them into modern frameworks remains an key open challenge toward efficient spatial reasoning.

The hippocampal-entorhinal (HC–EC) system is central to spatial and episodic memory(Buzsáki & Moser, 2013; Eichenbaum et al., 2007; Eichenbaum, 2017; Whittington et al., 2022). Experiments highlight two key computational principles: the functional dissociation between MEC and LEC pathways(Hargreaves et al., 2005; Knierim et al., 2014), and the hippocampus's role in binding structural and sensory information(Eichenbaum et al., 2007). These insights have motivated rich computational models with factorized structural and sensory representations(Hasselmo et al., 2002; Franzius et al., 2007; Bush et al., 2015), where MEC provides path-integration–based structural codes and the hippocampus binds them with LEC-derived sensory inputs. Building on this foundation, several learnable HC-EC–inspired models have been proposed, including CSCG(George et al., 2021; Raju et al., 2024), the Tolman–Eichenbaum Machine (TEM)(Whittington et al., 2020), and Vector-HaSH(Chandra et al., 2025). As shown in Fig. 1AB, TEM offers a unified computational framework that learns abstract structure in an unsupervised manner and generalizes this structure to novel environments, making it a promising basis for hippocampus-like memory models. However, existing HC–EC models focus mainly on simplified synthetic settings(Whittington et al., 2020; Raju et al., 2024; Zou et al., 2024; Chandra et al., 2025). How to integrate these properties with modern deep learning architectures and scale them to richer, real-world tasks remains an open challenge.

For example, the original TEM uses tensor-product Hebbian weights for relational memory, which is biologically plausible but capacity-limited. TEM-t replaces these with key-value memory and self-attention-based retrieval, improving efficiency but still incurring high computational cost(Whittington et al., 2021). Moreover, it inherits the constraints of transformer-based architectures, such as limited context windows. Furthermore, both models demand careful memory management and parameter tuning to realize novelty-based storage and retrieval. Together, these limitations hinder the practical integration of hippocampal-inspired spatial memory into modern deep learning, despite their intriguing conceptual motivation.

To address these challenges, we introduce mm-TEM, a more scalable and efficient hippocampus-inspired structural memory, and Hippoformer, a hybrid model that integrates mm-TEM with transformers. mm-TEM introduces a meta-MLP memory system, meta-trained for associative binding. Building on this, Hippoformer combines this mm-TEM with transformer, yielding complementary strengths. Despite their simplicity, both models achieve good performance on long-horizon prediction tasks in 2D and 3D environments. Our main contributions are:

1. **mm-TEM:** We propose an efficient and scalable TEM variant with a newly designed meta-MLP based relational memory. mm-TEM substantially improves training efficiency over TEM, generates grid-like patterns through self-supervised learning, and uncovers an intriguing link between prediction horizon and grid scales, offering new insights into how different spatial grid scales are formed at the implementation level.

2. **Systematic evaluation:** mm-TEM is extensively tested on long sequences, large-scale environments, and multi-step prediction. It generalizes significantly better than baselines such as transformers and Titans. Ablation studies illustrate the importance of the auxiliary relational loss, and further analyses show that its generalization stems from explicit understanding of spatial structures and rules, demonstrating mm-TEM as an effective structural spatial memory system.

3. **Hippoformer:** We propose Hippoformer, which integrates mm-TEM with a transformer to combine the structural spatial memory of mm-TEM with the precise working memory capability of Transformer. This synergy enhances generalization in both 2D and 3D prediction tasks, demonstrating the potential of hippocampal-inspired architectures in tackling complex domains.

In summary, mm-TEM provides an efficient and scalable structural spatial memory system. And when combined with Transformer, Hippoformer has an potential to serve as a building block for enhancing spatial reasoning in deep learning.

## 2 METHOD

In this section, we present the mm-TEM and Hippoformer architecture in detail. We use the 2D grid-world prediction task as an example, where an agent moves with discrete actions $(up, down, left, right)$(Whittington et al., 2020). The input sequence is denoted as $a_0, s_0, a_1, s_1 \ldots, a_t, s_t$, where $s_t \in \mathbb{R}^d$ is the sensory observation at time $t$ and $a_t$ a one-hot action. The model is trained to predict the next sensory obsevation $s_{t+1}$ given $a_{t+1}$, thereby mimicking hippocampal predictive coding during spatial exploration Whittington et al. (2022). The overall model structure is illustrated in Fig. 1.

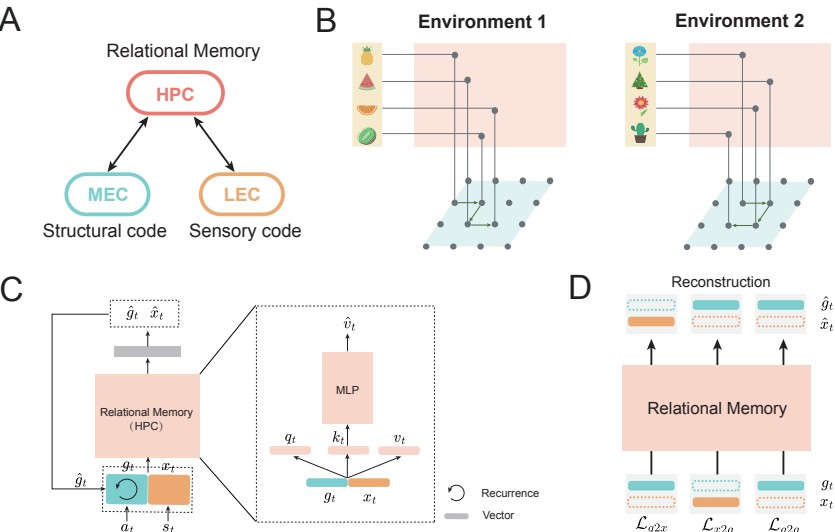

Figure 1: Factorization of structure and content in the hippocampus and model. (A) The hippocampal–entorhinal system functions as a memory system: MEC encodes structural information, LEC encodes sensory content, and HPC integrates both via conjunctive coding. (B) Structural codes in MEC can be reused across environments, enabling compositional generalization, adapted from Whittington et al. (2020). (C) The model comprises two components: a path integration network and a relational memory network, implemented as an meta-MLP memory. (D) The relational memory module is trained to reconstruct sensory codes from structural codes, structural codes from sensory codes, and both from joint inputs.

### 2.1 MODEL ARCHITECTURE AND TRAINING

Following previous modeling studies (Hasselmo et al., 2002; Franzius et al., 2007; Bush et al., 2015; Stachenfeld et al., 2017; Waniek, 2019; Rajan et al., 2016) and TEM computational framework (Whittington et al., 2020), mm-TEM consists of two key modules: a path integration network and a relational memory network (Fig. 1C). The path integration network receives action inputs $a_t$ and predicts the corresponding structural code $g_t$, while the relational memory network binds $g_t$ with the sensory code $x_t$, extracted from observations $s_t$ through a feature encoder. This design enables flexible bidirectional retrieval between structural and sensory domains.

**Path Integration Network**. Inspired by grid system in MEC(Moser et al., 2017), the network enforces basic spatial consistency rules (e.g., North + East + West + South = 0). Following Gao et al. (2021), we implement it as a two-layer MLP $f_g$ with ReLU activations to map the action $a_t \in \mathbb{R}^{d_a}$ to a transformation matrix $W_t^g \in \mathbb{R}^{d_g \times d_g}$:

$$W_t^g = f_g(a_t). \tag{1}$$

The structural code is then updated as

$$\tilde{g}_t = \text{ReLU}(W_t^g g_{t-1}), \quad g_t = \frac{\tilde{g}_t}{\|\tilde{g}_t\|_2}, \tag{2}$$

where $\ell_2$-normalization ensures that $g_t$ remains a unit vector.

**Relational Memory Network**. Mimicking hippocampal relational memory (Eichenbaum et al., 2007), the network binds structural and sensory codes into a joint representation $m_t = [g_t, x_t]$, enabling bidirectional retrieval between $g_t$ and $x_t$. To replace the computationally expensive Hebbian weights in TEM, we introduce a meta-MLP with hierarchical fast weights $\Theta_t$, inspired by Titans, to store relational knowledge. This design enables dynamic memorization, forgetting, and querying at test time, alleviating the complex memory management and parameter tuning required in TEM and TEM-t.

Concretely, the relational network first projects $m_t$ into three latent vectors:

$$k_t = W_k m_t, \quad v_t = W_v m_t, \quad q_t = W_q m_t, \tag{3}$$

where $k_t, v_t, q_t$ denote the key, value, and query representations, respectively. Rather than storing $m_t$ directly, the meta-MLP learns to associate key $k_t$ to value $v_t$ online by minimizing the reconstruction loss:

$$\mathcal{L}(k_t, v_t; \Theta_t) = \left\| f_{\mathrm{MLP}}(k_t; \Theta_t) - v_t \right\|_2^2, \tag{4}$$

where $f_{\mathrm{MLP}}(\cdot; \Theta_t)$ denotes the meta-MLP. The fast weights $\Theta_t$ are updated by incorporating prediction-error-driven adaptation and forgetting:

$$\Theta_t = (1 - \alpha_t)\Theta_{t-1} + H_t, \tag{5}$$

$$H_t = \eta_t H_{t-1} - \beta_t \nabla_\Theta \mathcal{L}(k_t, v_t; H_{t-1}). \tag{6}$$

Here, $\alpha_t \in [0, 1]$ is a data-dependent gating variable that controls forgetting, paralleling hippocampal mechanisms that decay less relevant memories to preserve capacity for novel ones(Benoit & Anderson, 2012; Liu et al., 2016). $\nabla_\Theta \mathcal{L}(\cdot)$ quantifies the gradient of prediction error - the degree of mismatch between the model's predictions and the input data - so that only unexpected inputs drive updates, akin to how hippocampus detects and prioritizes novel stimuli for long-term storage(Sinclair et al., 2021; Schomaker & Meeter, 2015). The term $\eta_t$ acts as a momentum factor, averaging prediction error over a tunable timescale to stabilize learning(Bittner et al., 2017), while $\beta_t$ is the learning rate. All parameters are derived from the input concatenation: $\alpha_t = \sigma(W_\alpha m_t)$, $\eta_t = \sigma(W_\eta m_t)$, $\beta_t = \sigma(W_\beta m_t)$, where $\sigma$ is the sigmoid function.

The query vector $q_t$ retrieves from memory via $f_{\mathrm{MLP}}(q_t; \Theta_t)$, and the retrieved representation is explained as a joint reconstruction $\hat{m}_t = [\hat{g}_t; \hat{x}_t]$.

To explicitly enforce relational binding, we introduce three auxiliary relational losses (Fig. 1D):

(1) Structure from content: retrieve $\hat{g}_t$ given only $x_t$ ($m_t = [\mathbf{0}, x_t]$), minimized by $\mathcal{L}_{x2g} = \|\hat{g}_t - g_t\|_2^2$.

(2) Structure from structure : retrieve $\bar{g}_t$ given only $g_t$ ($m_t = [g_t, \mathbf{0}]$), minimized by $\mathcal{L}_{g2g} = \|\bar{g}_t - g_t\|_2^2$.

(3) Content from the structure: retrieve $\hat{x}_t$ given only $g_t$ ($m_t = [g_t, \mathbf{0}]$), minimized by $\mathcal{L}_{g2x} = \|\hat{x}_t - x_t\|_2^2$.

The total relational loss is $\mathcal{L}_{\mathrm{rel}} = \mathcal{L}_{x2g} + \mathcal{L}_{g2g} + \mathcal{L}_{g2x}$. Since $\mathcal{L}_{g2x}$ will be accounted for in the main predictive-learning objective, it can be absorbed into that term. Thus, the relational loss can be simplified to $\mathcal{L}_{\mathrm{rel}} = \mathcal{L}_{x2g} + \mathcal{L}_{g2g}$.

Finally, mm-TEM incorporates a feedback loop from relational predictions to the path integration network, providing error correction during navigation (Fig. 1C; see Appendix. A.2 for details).

**mm-TEM Training**. The objective of mm-TEM is to predict the next observation given past sensory inputs and actions. The model is trained in a self-supervised manner. During training, in the relational memory network, the projection matrices $W_k, W_v, W_q$ are meta-trained in the outer optimization loop, while the connection weights of the meta-MLP are optimized in the inner optimization loop. During testing, the connection weights of the meta-MLP are updated online using gradient-based update rules. We introduce a hyperparameter

*mb* to control the memory update frequency in the relational memory network. Specifically, the connection weights in the meta-MLP are updated every *mb* steps. A larger *mb* results in sparser updates, which improves training efficiency but requires the model to rely on older information when predicting the next observation. Before downstream task training, we warm up the relational memory network through meta-training with random $\{a_t, s_t\}$ samples, utilizing only the relational loss. This procedure helps stabilize training. All networks are optimized using the Adam optimizer. Additional training details and the objective loss function are provided in Appendix. A.5. For all primary results, models are trained on sequence lengths of 128 or 256 using full backpropagation through time (BPTT) without truncation(See Fig. A.5 for ablation).

**Hippformer Architecture and Its Training**. Building on mm-TEM, we propose Hippoformer, which integrates mm-TEM and a one-layer Transformer in parallel to leverage the complementary strengths of both modules. In Hippoformer, the mm-TEM component is also warm-started following the same protocol described above. Rather than receiving the path integration network's output $g_t$, the Transformer architecture here takes an action embedding generated by a single-layer MLP (Kitaev et al., 2020). This embedding is concatenated with $a_t$ and $s_t$ to form the final input. The model is trained using the Adam optimizer. Additional architectural and training details are provided in Appendix. A.5.

**Control Models**. We used Transformer and Titans as control models. They receive identical input tokens to those used by the the transformer module in Hippoformer. Unless otherwise stated, all models are trained and evaluated under the same task setting for fair comparison.

## 3 RESULTS

### 3.1 EFFICIENT TRAINING AND EMERGE OF GRID-LIKE REPRESENTATIONS IN MM-TEM

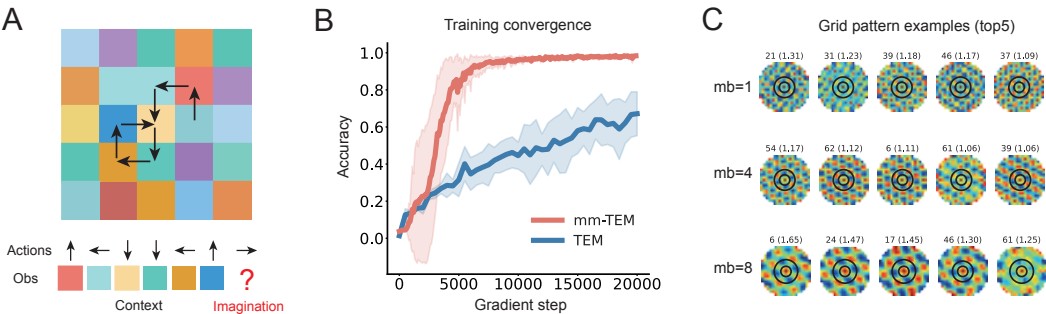

Figure 2: Task schematic and network performance. (A) A 2D Grid prediction task (example 6×6). The network predicts the next observation based on the current action at each step in sequence. (B) Multi-step test accuracy over gradient steps for mm-TEM vs TEM (same batch/sequence training length), averaged over 4 seeds. To assess generalization, the networks predict till 256 steps from an initial 64-step context, with results averaged over four trials. More training details and parameters, see Appendix. A.1, A.2 and A.5. (C) In the path integration network, emerged grid scale varies with hyperparameter *mb*. Five top-gridness neurons shown per condition (more examples in Figs. A.12, A.13 and A.14).

We first ask whether mm-TEM can efficiently solve spatial reasoning task and acquire MEC-like representation. To test this, we evaluate the model on 2D grid prediction tasks (Fig. 2A)(Whittington et al., 2020), where the agent must predict the next observation based on the current action at each time step within a 256-step sequence. For each trial, the environment is sized between 9x9 and 11x11 with randomized observations, requiring the network to infer the underlying spatial structure and rules to generalize effectively. Since observations at unseen locations within this discrete environment are unpredictable, both training and evaluation are confined to positions previously encountered within each sequence. More evaluation details, see Appendix. A.4.

In terms of spatial reasoning performance, mm-TEM reaches nearly 90% test accuracy within only 5,000 gradient steps, while TEM converges very slowly, achieving only about 60% accuracy even after 20,000 steps (Fig. 2B), highlighting the superior training efficiency of mm-TEM.

In terms of internal representations, analysis of the path-integration network of mm-TEM further reveals periodic grid-like representations (Fig. 2C). Notably, the grid scale is directly modulated by the update-frequency hyperparameter $mb$: larger $mb$ yields coarser grids, whereas smaller $mb$ produces finer scales. Since $mb$ sets the effective prediction horizon, This suggests a novel mechanism for grid-scale diversity in MEC (Fyhn et al., 2004) at the implementation level, and further demonstrates that diverse grid scales can naturally arise from multi-timescale predictions in the brain Waniek (2019); Stachenfeld et al. (2017); Dordek et al. (2016). Overall, mm-TEM not only trains efficiently but also reproduces grid-like patterns in the HC–MEC system, offering new insights into the computational basis of grid-scale diversity.

## 3.2 GENERALIZATION OF MM-TEM IN 2D GRID PREDICTION TASKS

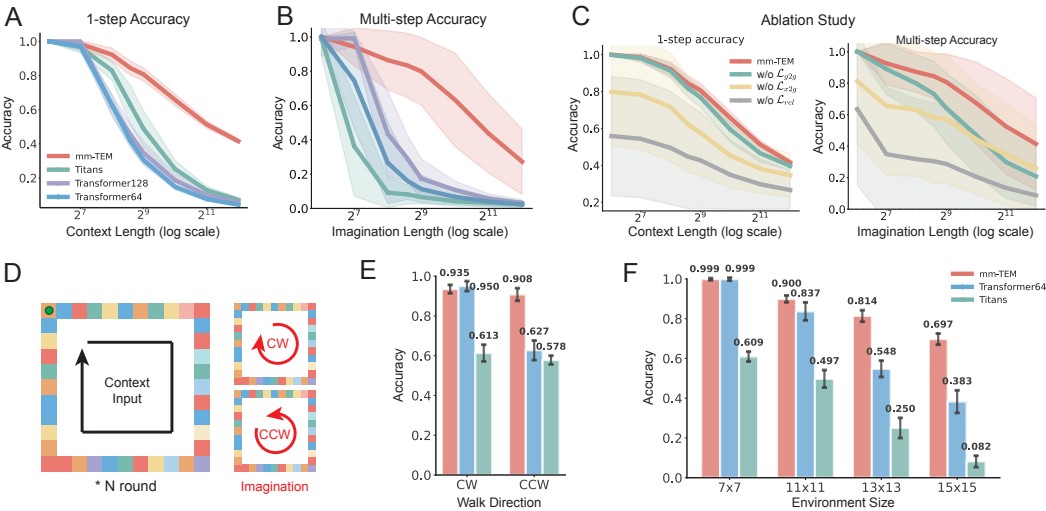

Figure 3: The Generalization and Ablation of mm-TEM. We compare mm-TEM with Titans and Transformer baselines on 2D grid prediction tasks (training length = 128). Titans uses a 3-layer MAC architecture. Transformer128 and Transformer64 denote 3-layer transformers with window sizes of 128 and 64 steps, respectively. (A) 1-step prediction accuracy vs context length. Networks receive an action–observation context sequence and predict the next observation. (B) Multi-step imagination accuracy vs imagination length. Networks observe a fixed 64-step context, then generate future observations conditioned on varying action lengths. (C) Ablation of auxiliary relational loss. "w/o g2g" removes $\mathcal{L}_{g2g}$, "w/o s2g" removes $\mathcal{L}_{s2g}$, "w/o rel" removes all auxiliary relational memory losses. (D) Circular-grid test setup. Networks explore an 11×11 circular environment clockwise for context, then imagine trajectories in clockwise or counterclockwise directions. (E) Clockwise vs counterclockwise performance across different architectures. (F) Effect of environment size, ranging from 7×7 to 15×15. All results are averaged over 3 seeds (see Appendix. A.3, A.5, Fig. A.3, Fig. A.4 and Fig. A.11 for more details).

Mimicking the HC–EC system, mm-TEM acts as a structured memory that organizes knowledge for generalization. We ask: how well does such a system generalize compared to modern architectures like Transformers and Titans? To answer this, we systematically evaluate mm-TEM against these baselines in diverse settings. In the one-step imagination setting, models explore environments with varying context lengths and predict the next observation. In the multi-step imagination setting, models receive a fixed 64-step context and predict fu-

ture observations conditioned on action sequences of varying lengths. These tasks probe mm-TEM's ability to generalize beyond its training horizon.

In the one-step prediction task (Fig. 3A), all models perform well within the 128-step training horizon. However, Transformer and Titans rapidly degrades once the context length extends beyond this range. In contrast, mm-TEM maintains more robust performance even with sequences up to 4096 steps, retaining 40% accuracy where baselines collapse, highlighting its long-term generalization ability.

In the multi-step imagination task (Fig. 3B), the Transformer model with a 128-step window performs almost perfectly within its training range, but quickly drops off outside it, suggesting reliance on sequence memorization. Titans show similar behavior. In contrast, mm-TEM maintains relatively robust long-term performance, suggesting that it has grasped the underlying spatial structure for generalization.

To determine the role of auxiliary relational loss (Fig. 3C) on this ability, we conduct ablations. Removing either $\mathcal{L}_{g2g}$ or $\mathcal{L}_{s2g}$ significantly reduces generalization ability, and eliminating all relational terms leads to severe performance degradation, confirming their importance.

Moreover, we further probe generalization under distribution shifts. In the circular-grid test (Fig. 3DE), mm-TEM achieves over 90% accuracy in the challenging counterclockwise condition, while Titans and Transformer suffer accuracy drops by up to 30%, underscoring mm-TEM's superior spatial reasoning ability. When scaling environment size from $7 \times 7$ to $15 \times 15$ (Fig. 3F) without additional training, all models decline, but mm-TEM deteriorates much more slowly and consistently outperforms the baselines. These results further show that mm-TEM generalizes beyond its training horizon, and captures spatial structure and rules more faithfully than control models, which appear to primarily rely on rote memorization.

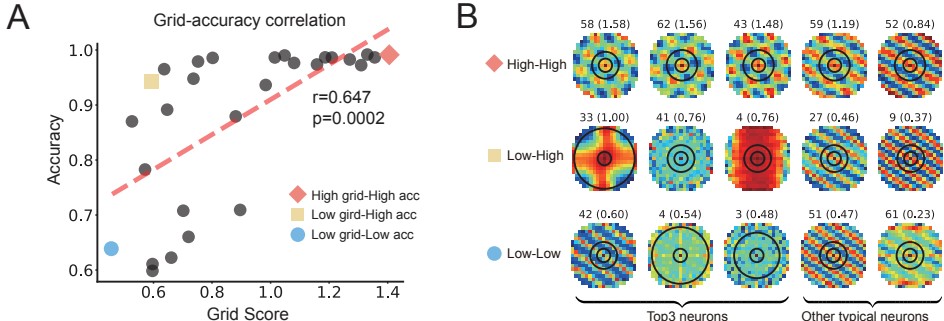

Figure 4: Long-horizon generalization and grid representations. (A) Multi-step prediction accuracy (imagination length = 512) positively correlates with grid score in mm-TEM path integration networks (r = 0.647, p = 0.0002), indicating that stronger grid-like regularity supports better generalization. (B) Representative models with high–high, low–high, and low–low grid–accuracy combinations show distinct autocorrelation patterns. For each model, the three neurons with the highest grid scores ("Top3 neurons") and other typical neurons are displayed, highlighting differences in grid-cell regularity across models. More details are provided in Appendix. A.5, Fig. A.8, Fig. A.15, A.16 and A.17.

To uncover why mm-TEM exhibits good generalization in long-horizon inference, we examine the relationship between a model's grid score and its multi-step imagination accuracy. As shown in Fig. 4A, multi-step generalization performance in mm-TEM is closely tied to the quality of its grid-like representations.Models with higher grid scores in the path-integration network consistently achieve higher prediction accuracy (further analysis provided in Appendix. A.8), indicating a positive correlation between grid score and prediction accuracy.

Interestingly, we also observe cases where models with relatively low grid scores still achieve relatively high accuracy. Visualization (Fig. 4B) reveals that these models develop alterna-

tive - but still regular - neural representations, in contrast with the unitary, unstructured patterns found in models with both low grid scores and low accuracy.

Taken together, these results suggest that the presence of strongly grid-like cells may reflect effective structure learning, thereby facilitating long-horizon generalization.

### 3.3 HIPPOFORMER BENEFITS FROM SHORT- AND LONG-TERM MEMORY INTEGRATION

From a memory perspective, Transformer with limited window size functions as precise short-term memory through accurate key–value caching, while mm-TEM provides a structured but less precise long-term memory. To leverage their complementary strengths, we propose Hippoformer, a unified architecture that combines a one-layer Transformer with mm-TEM. Both modules process the input embeddings independently, and their outputs are concatenated and integrated by an MLP (Fig. 5A). We evaluate all models on the 2D grid prediction task.

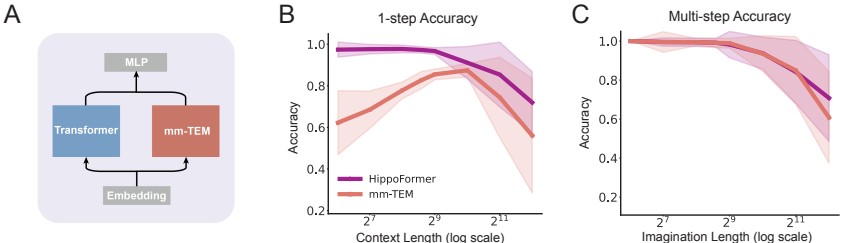

Figure 5: Hippoformer architecture and generalization in 2D grid prediction. Both Hippoformer and mm-TEM are trained using $mb = 8$ and 256-step sequences. (A) Hippoformer combines a one-layer Transformer and mm-TEM, both receiving action and sensory embeddings; their outputs are concatenated and integrated by an MLP. (B) One-step prediction accuracy of Hippoformer and mm-TEM across different context lengths. (C) Multi-step imagination accuracy across different imagination lengths, comparing Hippoformer and mm-TEM. Additional tests for effects of the memory update frequency $mb$ are provided in Appendix. A.6 and Fig. A.2

As shown in Fig. 5B, mm-TEM with $mb = 8$ can be trained efficiently, but its one-step prediction drops at short context lengths due to limited access to recent information, requiring longer contexts to reach good performance. When combined with a Transformer, however, Hippoformer generalizes across both short and long context lengths. In the multi-step imagination task (Fig. 5C), where performance depends primarily on the mm-TEM component, both models achieve similar accuracy with no significant difference.

Overall, Hippoformer successfully integrates the strengths of both memory systems. The Transformer provides short-term memory for accurate short-range prediction, while mm-TEM supports structured long-horizon forecasting. This hybrid design is appealing for applications, as reducing MLP memory update frequency in mm-TEM greatly improves training efficiency and minimizes redundant memory storage, though at the cost of short-term accuracy (see Appendix. A.6 and Fig. A.2). Consequently, Hippoformer achieves both efficient training and good generalization across diverse temporal horizons.

### 3.4 HIPPOFORMER LEVERAGES THE SYNERGY BETWEEN ABSTRACTION AND MEMORIZATION

The hippocampus supports not only memorization but also abstraction, whereas traditional TEM and TEM-t models primarily emphasize memory storage and memory-based inference. In contrast, mm-TEM moves beyond structured memorization by enabling abstraction through parametric relational memory. Hippoformer merges Transformers' short-term memory with mm-TEM's long-horizon abstraction to boost generalization. To evaluate this capacity, we design a 3D empty environment task (as shown in Fig. 6AB) (Pasukonis et al., 2022). In this new setting, layout textures and egocentric trajectories are randomly sampled. Observation features are extracted through an encoder, concatenated with action

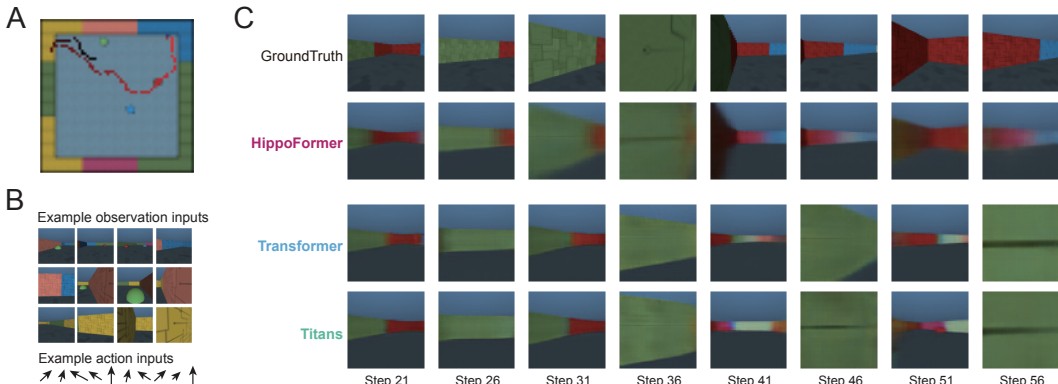

Figure 6: Hippoformer generalization in 3D environment prediction tasks. (A) Example 3D environment with randomly sampled layouts and navigation trajectories. (B) Example trajectory showing sequences of egocentric observations and actions. (C) Visualization of imagined trajectories from different models, with snapshots shown every 5 steps. More details see Appendix. A.9 and Fig. A.18 and A.19.

inputs, and fed into the models. Each model is trained to predict the next egocentric frame over 64-step sequences. They are then evaluated on both one-step and multi-step prediction, in a manner similar to the 2D grid experiments. Performance is measured using the reconstruction error of predictions in pixel space (see Appendix. A.4 for details).

Table 1: Performance comparison of different models on prediction error in 3D environments. The results are shown for both one-step and multi-step conditions, with errors reported in units of $1e^{-3}$. The results are averaged over 3 seeds.

| Models | 1-step Prediction Error (1e-3) | | | m-step Imagination Error (1e-3) | | |
|---|---|---|---|---|---|---|
| | Full | Visible | Not Visible | Full | Visible | Not Visible |
| Transformer | 1.29±0.00 | 0.67±0.00 | 2.15±0.00 | 36.13±5.0 | 11.49±3.3 | 38.07±13 |
| Titans | 1.32±0.00 | 0.69±0.00 | 2.20±0.05 | 33.42±4.6 | 10.60±2.4 | 35.21±13 |
| mm-TEM | 5.10±0.05 | 4.23±0.05 | 6.53±0.05 | 14.30±0.36 | 13.23±3.05 | 14.40±0.16 |
| Hippoformer | **1.27±0.00** | **0.67±0.00** | **2.09±0.05** | **9.71±0.04** | **2.72±0.01** | **10.27±3.6** |

We systematically evaluate Hippoformer using three complementary metrics: error across the entire sequence ("full"), error on visible frames ("visible"), and error on non-visible frames (see Appendix. A.4 for details). Visible frames probe the model's ability to exploit structured memory, whereas non-visible frames require abstraction from historical context. As shown in Tab. 1, Hippoformer outperforms both transformer and Titans models slightly in one-step prediction but markedly in multi-step imagination. Consistent with these findings, Fig. 5C shows that Hippoformer maintains coherent predictions over long horizons in multi-step settings, whereas transformer and Titans models exhibit oscillatory errors around 36–56 steps, appearing to stack over time. Notably, for consistency with the Hippoformer, the standalone mm-TEM is also trained with $mb = 8$ on the 3D task. A large $mb$ inherently biases mm-TEM toward long-horizon prediction—enhancing multi-step imagination but reducing single-step accuracy (Tab. 1), a limitation that is compensated by the Transformer component in Hippoformer. Additional results are provided in Appendix A.9. Overall, these results demonstrate that Hippoformer effectively leverage abstraction and memorization, with its two modules cooperating to achieve robust long-term prediction.

## 4 DISCUSSIONS

In this work, we introduce mm-TEM and Hippoformer, two hippocampus-inspired models for prediction and spatial reasoning. mm-TEM trains more efficiently than standard TEM and spontaneously develops grid-like codes, whose grid scale is modulated by the prediction horizon, offering a new perspective on grid diversity at the implementation level. Additionally, we propose Hippoformer, which integrates Transformers and mm-TEM. A natural division of labor emerges: Transformers primarily capture short-term dependencies, while

mm-TEM supports long-horizon forecasting through robust grid codes in 2D environments. In 3D environments, Transformers contribute to short-term memorization, whereas mm-TEM focuses on long-term abstraction. Together, these complementary roles yield both improved training efficiency and better generalization.

**Related works**. Our work extends the computational framework of the HC-EC system. Existing models, such as CSCG (George et al., 2021), Vector-HaSh(Chandra et al., 2025), TEM(Whittington et al., 2021), and TEM-t(Whittington et al., 2021), are conceptually elegant but face limitations in scaling to modern deep learning architectures. For example, TEM relies on computationally expensive tensor-product Hebbian memory; TEM-t is constrained by transformer window size and requires complex memory updates; and Vector-HaSh is non-differentiable. These limitations hinder their application to complex tasks. In contrast, we propose mm-TEM, which employs a hierarchical MLP as a relational memory system. Augmented with auxiliary relational losses, mm-TEM offers a powerful, flexible memory mechanism that integrates seamlessly with modern transformers, enabling its use in more complex environments.

Long-sequence modeling is a central challenge in machine learning. Recent architectures such as Mamba(Gu & Dao, 2023), Titans(Behrouz et al., 2024), and Gated Delta Networks(Yang et al., 2024) represent important advances through structural initialization, hierarchical MLP memory, and novelty-based Hebbian rules. However, real-world information is inherently spatiotemporal, and simply enlarging memory capacity while ignoring its underlying structure is an inefficient strategy. To address this, we introduce Hippoformer, a novel hybrid memory system that combines the precise short-term memory of transformers with the structured long-term memory of mm-TEM. This design enables more efficient organization of memory sequences, making Hippoformer a promising architecture for long-sequence modeling.

It has long been proposed that diverse grid scales may be a natural consequence of multi-timescale predictions in the brain. For example, Waniek analytically derived diverse grid scales from multi-scale predictions (Waniek, 2019). Similarly, Stachenfeld et al. and Dordek et al. both proposed that diverse grid scales can arise as basis functions of multiscale place codes (Stachenfeld et al., 2017; Dordek et al., 2016). In contrast to these works, mm-TEM produces different grid scales emergently through self-supervised learning, achieved simply by adjusting the memory-update frequency, without imposing any prior multiscale place structure. This end-to-end emergence is important because it allows prior theoretical ideas about multiscale grid representations to be tested and extended in more complex and realistic task settings.

Additionally, mm-TEM offers a new perspective on the biophysical mechanisms underlying diverse grid scales in the brain. Unlike prior approaches that learn to reconstruct predefined multiscale place representations (Stachenfeld et al., 2017; Dordek et al., 2016), mm-TEM suggests that memory-update frequency itself may play a critical role by implicitly implementing a multiscale prediction horizon. Biologically, variation in memory-update frequency may be able to related to known gradients along the ventral–dorsal hippocampal axis, such as oscillation-frequency gradients (Goyal et al., 2020) or receptor-expression gradients (Strange et al., 2014).

**Limitations and Future work**. While mm-TEM provides an efficient structured memory system, our current Hippoformer design only illustrates a straightforward parallelization of transformer and mm-TEM. Moreover, the present Hippoformer is limited to a single-layer design, without leveraging the model and computation scaling that has been shown to be crucial in large language models(Kaplan et al., 2020).

Future work should investigate more efficient integration schemes and multi-layer scaling, positioning mm-TEM as a scalable fundamental building block for large systems and spatial reasoning tasks. More broadly, mm-TEM's simplicity may enable hierarchical models of the hippocampus, offering a computational handle on how biological dorsal–ventral representational gradients give rise to functional specialization(Strange et al., 2014; Maurer & Nadel, 2021).

ACKNOWLEDGMENTS

This work was supported by the National Natural Science Foundation of China (62336007, B. Hong).

ETHICS STATEMENT

This work does not involve sensitive datasets, human subjects, or potentially harmful applications. Therefore, we have not identified any obvious ethical concerns.

REPRODUCIBILITY STATEMENT

The paper and appendix provide detailed descriptions of model architectures, hyperparameters, training procedures, data preprocessing steps, and computational resource configurations. We also include ablation studies and additional results to ensure that the main conclusions are robust.

THE USE OF LARGE LANGUAGE MODELS (LLMs)

We used large language models (LLMs) for minor text polishing, but not for generating scientific content, experiments, or analysis. In addition, we employed LLMs to refine visualization code for clarity and readability; these edits did not affect any experimental results.

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

# A   APPENDIX

## A.1   DETAILS OF 2D AND 3D PREDICTION TASKS

**2D grid prediction task**. In the 2D grid environment, the agent can take four discrete actions (*up, left, right, down*), which are encoded as one-hot vectors. Observations are sampled from a uniform distribution over 64 sensory objects, and in each trial, the observations are drawn independently from this set. Since the environment is discrete and sensory observations are uncorrelated, generalization in this setting primarily depends on structural memory-based inference rather than feature abstraction. Consequently, predicting observations at unvisited locations is not meaningful. For all 2D grid prediction tasks, the models are trained on grid environments of sizes $11 \times 11$.

**3D environment prediction task**. The 3D environment is built on MemoryMaze3D (Pasukonis et al., 2022), with the layout simplified to an empty 2D plane. Environment textures are randomly sampled in each trial. Unlike the allocentric observations in the 2D grid environment, the observations here are egocentric view images, which are inherently more complex. The action space consists of a discrete set of actions, including moving forward, turning left, turning right, staying still, and combinations of moving forward with turning. However, the resulting movements are continuous, subject to noise and acceleration. In this environment, unvisited observations can be inferred from nearby spatial information, making feature abstraction a critical factor in the 3D environment prediction tasks.

## A.2   DETAILS OF MM-TEM ARCHITECTURE

The mm-TEM architecture primarily consists of a path integration network and a relational memory network. Unlike the relational memory in TEM and TEM-t, which incorporate hand-designed relational priors - TEM uses a tensor-product-based Hebbian mechanism, and TEM-t employs key-value pairs - our relational memory does not rely on any manually injected priors. Instead, we self-supervise the network with a simple auxiliary relational loss, allowing the memory system to learn relational structure autonomously. This approach makes the memory system more flexible and scalable compared to those in TEM and TEM-t.

mm-TEM receives both action vectors and sensory feature vectors as inputs. In the 2D grid environment, sensory features are extracted using a 2-layer MLP encoder, whereas in the 3D environment they are extracted using a 4-layer convolutional neural network and 1-layer MLP.

For the relational memory network, we follow the standard configuration used in Titans memory. Specifically, we employ a 3-layer MLP as the core of the meta-MLP network. The relational memory module processes the input using two attention heads, each with 64 dimensions. The meta-memory weights are updated using a 2-order momentum rule. These settings are kept identical across both 2D and 3D environments, and are also matched to the Titans baseline for fair comparison.

The path integration network processes the action sequences and generates structural codes. we use 64 grid cells for 2D environments and 128 grid cells for 3D environments. Due to noise-driven error accumulation, the path integration network requires correction signals from sensory observations to recalibrate the structural codes, analogous to the HC–EC system in biological brain. Further error correction details are provided below.

**Error correction in the path integration system.** In the HC–EC system, the HC provides relational or spatial feedback - likely in the form of conjunctive codes that bind spatial and sensory information - which can help recalibrate MEC structural representations (Diehl et al., 2018; Mulas et al., 2016). To emulate this mechanism, we introduce a feedback loop from the relational memory network. Specifically, given an action $a_t$, the path integration network generates a structural code $g_{gen,t}$. Together with the sensory code $x_t$, a memory query $m_t = [g_{gen,t}; x_t]$ is formed and sent to the relational memory network, which retrieves a structural code $\hat{g}_{gen,t}$ and a content code $\hat{x}_t$. The retrieved code is then fed back into the path integration network and combined with $g_{gen,t}$ as follows:

$$g_{inf,t} = g_{gen,t} + \alpha(\hat{g}_{gen,t} - g_{gen,t}) \cdot f_{delta}(g_{gen,t}, \hat{g}_{gen,t}, sg(||x_t - \hat{x}_t||^2))$$

where $f_{delta}(\cdot)$ is two-layer MLP that predict the variance of the integrated structural code, respectively. $sg(.)$ means stop gradients. The scalar $\alpha$ controls the integration ratio, which slowly increases in the first 2000 gradient steps. The updated structural code, $g_{inf,t}$, then becomes the new state of the path integration network and is stored in the relational memory alongside the sensory code, $x_t$.

In mm-TEM, the meta-MLP memory is updated once every $mb$ time steps. At each time $t$, the model aggregates the recent pairs $(g_{gen,t}, s_t)$ collected since the last update. Given an input sequence $[\ldots, (g_{gen,t}, s_t), (g_{gen,t+1}, s_{t+1}), \ldots, (g_{gen,t+mb}, s_{t+mb}), \ldots]$, if the previous memory update occurs at time $t$, the next one occurs at $t + mb$.

At time $t + mb$, the path integration network is first corrected using $(g_{gen,t+mb}, s_{t+mb})$. This pair is sent to the relational memory (mm-MLP), which retrieves a structural code $\hat{g}_{gen,t+mb}$. The retrieved code is then combined with the previous structural estimate $g_{gen,t}$ to produce an updated, corrected structural code $g_{inf,t+mb}$. After this correction step, the mm-MLP memory is then updated, using the batch $[(g_{gen,t}, s_t), \ldots, (g_{gen,t+mb-1}, s_{t+mb-1}), (g_{inf,t+mb}, s_{t+mb})]$.

Importantly, throughout the interval $t \to t+mb$, the network has no access to these "future" $(g, s)$ pairs. All predictions must rely solely on information available at and before time $t$. Consequently, the model must implicitly predict 1 to $mb$ steps ahead, and a larger $mb$ increases the effective prediction horizon.

**Objective Functions and Training.** The overall prediction and update process in mm-TEM is summarized as follows:

1. Given the action $a_t$, the path integration network computes the structural code $g_{gen,t}$.

2. With the input $m_t = [g_{gen,t}; \mathbf{0}]$, the relational memory network retrieves the sensory code, which is further decoded into the observation $\hat{s}_{gen,t}$. This yields the *generative prediction loss*:
$$\mathcal{L}_{\text{gen}} = \|\hat{s}_{gen,t} - s_t\|_2^2.$$

3. Using the joint structural–sensory code $m_t = [g_{gen,t}; x_t]$, the relational memory network predicts the corrected structural code $\hat{g}_{gen,t}$. Combining the generative and feedback-retrieved structural codes $(g_{gen,t}, \hat{g}_{gen,t})$, mm-TEM produces the corrected structural code $g_{inf,t}$ using the above error correction process. This gives the *consistency loss*:
$$\mathcal{L}_{\text{con}} = \|g_{gen,t} - g_{inf,t}\|_2^2.$$

4. The corrected structural code $g_{inf,t}$ is then passed again to the relational memory network with $m_t = [g_{inf,t}; \mathbf{0}]$. The network predicts the sensory code $\hat{x}_{inf,t}$, which is decoded into the observation $\hat{s}_{inf,t}$. This defines the *inference prediction loss*:
$$\mathcal{L}_{\text{inf}} = \|\hat{s}_{inf,t} - s_t\|_2^2.$$

5. Then, the integrated structural code $\hat{g}_{inf,t}$ and the sensory code $x_t$ are stored in the relational memory network using the online gradient-based update rule.

The main objective of mm-TEM is to predict the next observation given past sensory inputs and actions. It is trained via self-supervised manner. The total loss combines the predictive loss, consistency loss and relational loss:

$$\mathcal{L} = \gamma_{rel}\mathcal{L}_{\text{rel}} + \gamma_{gen}\mathcal{L}_{\text{gen}} + \gamma_{con}\mathcal{L}_{\text{con}} + \gamma_{inf}\mathcal{L}_{\text{inf}}, \tag{7}$$

where $\gamma_{rel}, \gamma_{gen}, \gamma_{con}$ and $\gamma_{inf}$ are scale factors of different losses. We train the entire mm-TEM end-to-end on the prediction task by minimizing $\mathcal{L}$. Optimization uses the Adam optimizer with a learning rate of 0.001. We apply a StepLR schedule with step size 500 and decay factor $\gamma = 0.9$. The warm-up phase lasts for 5,000 steps, and training runs for up to 20,000 steps.

When combined with Transformer, the model architecture of Hippoformer is shown in a more detaied version of Fig. 2A in Fig. A.1.

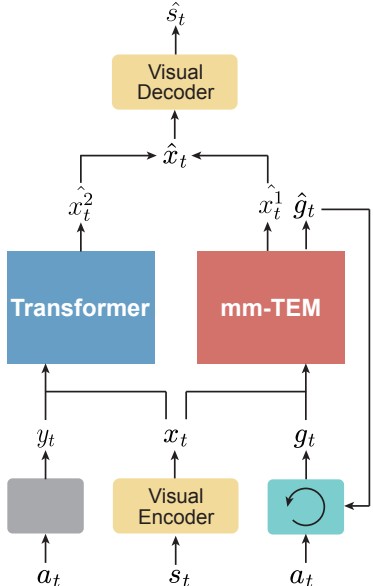

Figure A.1: Model architecture of Hippoformer, which integrates mm-TEM with a Transformer in parallel. Related to Fig. 2A

### A.3 CONTROL MODEL ARCHITECTURE

We compare mm-TEM and Hippoformer with two control architectures: Transformers and Titans. The model configurations are summarized as follows.

For the Transformer baseline, we employ a three-layer Transformer with two different temporal windows: 64 steps and 128 steps. The number of attention heads is set to $N_{\text{head}}^T = 8$ and the hidden dimension to $N_{\text{dim}}^T = 128$. Although the Transformer with a 64-step window has limited temporal context, its hierarchical three-layer structure still allows it to capture dependencies across the entire sequence.

For Titans, we adopt the MAC architecture, which incorporates an external memory component as contextual information. The meta-MLP parameters keep identical with our relational memory modules. Its Transformer component uses the same parameterization as the standard 64 windows Transformer. Both the Transformer and Titans share the same input layer, which embeds actions and sensory inputs into $N_{act} = 192$, $N_{sense} = 192$ in 2D environments, and $N_{act} = 128$, $N_{sense} = 4096$ in 3D environments.

In Hippoformer, the short-term memory module is implemented as a one-layer Transformer with a window size of 32. In both model(mm-TEM and Hippoformer), the input layer embeds grid representations into $N_{grid} = 64$, $N_{sense} = 64$ in 2D environments, and $N_{grid} = 128$, $N_{sense} = 4096$ in 3D environments.

### A.4 MODEL EVALUATION

In 2D environments, we use one-step prediction accuracy and multi-step imagination accuracy for evaluation, following TEM and TEM-t.

For one-step prediction, given a context sequence of length $T_1$ with state-action inputs $[(s_1, a_1), (s_2, a_2), \ldots, (s_{T_1}, a_{T_1})]$, the model generates next-state predictions $\hat{S} = [\hat{s}_2, \hat{s}_3, \ldots, \hat{s}_{T_1}]$. A predicted state is counted as valid only when its corresponding ground-truth was visited before $t$.

For multi-step imagination, after receiving a context of length $T_1$, the network performs $T_2$-step imagination. During imagination, an action sequence $[a_{T_1}, \ldots, a_{T_1+T_2}]$ is provided,

and the network produces predicted states $\hat{S} = [\hat{s}_{T_1+1}, \hat{s}_{T_1+2}, \ldots, \hat{s}_{T_1+T_2}]$. A predicted state is counted as valid only when its corresponding ground-truth was visited during the context period. For both one-step and multi-step, all accuracies are computed as the fraction of correct predictions among these valid predictions. Since unvisited states are inherently unpredictable in the 2D disctrete grid state setting, "generalization" is over unseen transitions (edges) rather than unseen states.

In 3D environments, the error metric is defined as the pixel-wise mean squared error normalized over all pixels. The definitions of "one-step" and "multi-step" imagination are similar to those in 2D. Our 3D environment has a size of $9 \times 9$. For evaluation, we roughly discretize the environment into a $9 \times 9$ spatial grid. Since the network receives egocentric observations, we further discretize the orientation into 12 bins, resulting in $9 \times 9 \times 12 = 972$ distinct states. If an observation falls into a spatial grid and orientation bin that has been encountered previously, we classify it as *visible*; otherwise, it is classified as *non-visible*. Note that this discretization is used *only during evaluation*.

A.5    TRAINING AND PARAMETER DETAILS

All models are implemented in `PyTorch` and trained on NVIDIA A100 GPUs. The training and parameter configurations corresponding to each figure are summarized below.

For the results in Fig. 2, we report the sequence length, learning rate, batch size, memory update frequency ($mb$), and testing environment size. Detailed settings are provided in Table A.1.

Table A.1: Training and parameter details for different experimental results.

| Figure | Sequence Length | Learning Rate | Batch Size | $mb$ | Env. Size |
|---|---|---|---|---|---|
| Fig. 2 | 256 | 1e-3 | 16 | 8 | 8-11 |
| Fig. 3 | 128 | 1e-3 | 16 | 1 | 8-11 |
| Fig. 4 | 256 | 1e-3 | 16 | 1/4/8 | 8-11 |
| Fig. 5 | 256 | 1e-3 | 16 | 8 | 8-11 |
| Fig. 6 | 64 | 5e-4 | 16 | 8 | 9 |
| Fig. A.11 | 128 | 1e-3 | 16 | 1 | 8-11 |
| Fig. A.2 | 128 | 1e-3 | 16 | 1/4/8 | 8-11 |

A.6    EFFECTS OF HYPERPARAMETER $mb$ ON MM-TEM

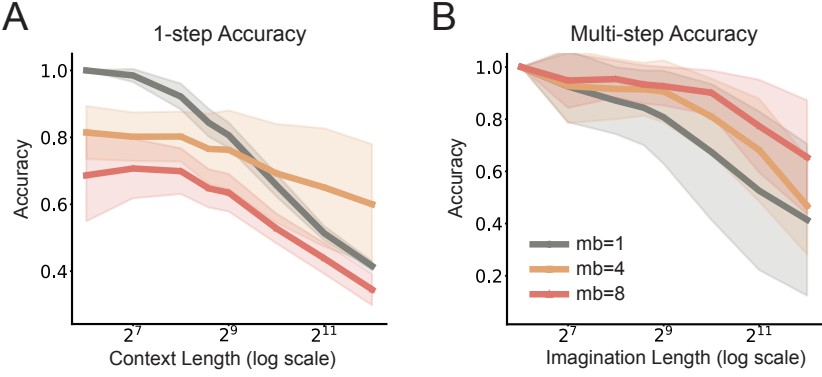

Figure A.2: Effects of Memory Update Frequency ($mb$) on Generalization Capacity of mm-TEM. We investigate how the memory update frequency parameter ($mb$) affects the generalization capacity of mm-TEM. All models are trained with a sequence length of 128. The generalization performance is evaluated under both one-step prediction (A) and multi-step prediction settings (B).

In mm-TEM, the meta-MLP memory is updated every $mb$ steps; thus, the hyperparameter $mb$ controls the memory update frequency. With a larger $mb$, the network must rely on information from $mb$ steps ago to predict the next observation, effectively increasing the prediction horizon. Therefore, $mb$ also balance short- and long-range predcitions. As shown in Fig. A.2, under one-step prediction, smaller $mb$ values emphasize short-range predictions but perform worse in multi-step prediction.

Ideally, models should perform well in both short- and long-range prediction and generalization. This is where Hippoformer demonstrates its advantage: by combining short-term memory from the Transformer with a limited window size and structured long-term memory from mm-TEM, Hippoformer achieves good performance across both short- and long-range predictions, as shown in Fig. 5 in the main text. Furthermore, larger $mb$ also improves training efficiency for mm-TEM. Taken together, this simple combination leverages the complementary strengths of both modules while also achieving high efficiency.

A.7 EFFECTS OF DIFFERENT POSITION ENCODING METHODS ON THE MODEL CAPACITY.

To determine whether mm-TEM and Hippoformer's advantage stems from hippocampus-like computational mechanism, we performed additional experiments and trained Transformer/Titan baselines with three PE variants: (1) Sinusoidal PE, which is static and absolute; (2) Rotary PE, which is a relative position embedding; (3) Dynamic PE, implemented by a path integration network identical to mm-TEM's PI module, serving as a powerful recurrent positional embedding. We evaluated both one-step and multi-step prediction and also tested transfer in the circular-grid environment (same as Fig. 3).

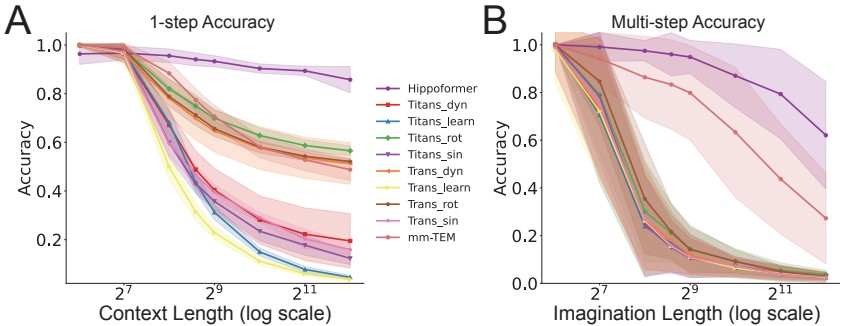

Figure A.3: Comparison of Model Generalization Using Diverse Positional Encodings. Titans and Transformers are evaluated with diverse PEs, including: absolute sinusoidal (e.g., Transformer_sin), rotary (e.g., Transformer_rot), Dynamic PE (e.g., Transformer_dyn), and the learned axial baseline PE used in the main text (e.g., Transformer_learn). All models are trained in 2D grid prediction tasks (training length = 128) as that in Fig. 3.(A) 1-step prediction accuracy vs context length. (B) Multi-step imagination accuracy vs imagination length. All results are averaged over 3 seeds.

Table A.2: Model Parameter Comparison for 2D Grid Tasks.

| Model | Parameters (M) |
|---|---|
| Titans | 30.62 |
| Trans | 29.63 |
| mm-TEM | 9.11 |
| Hippoformer | 10.06 |

As shown in Fig. A.3 and Fig. A.4, our key findings are as follows. (1) Rotary PE and dynamic PE substantially improve one-step prediction, in some cases reaching performance comparable to mm-TEM with $mb = 1$. (2) However, none of these PE variants allow Transformers or Titans to approach mm-TEM's performance on multi-step prediction or circular-grid transfer, both of which require robust long-range relational inference. (3) Hippoformer (Transformer + mm-TEM with $mb = 8$) consistently outperforms all PE baselines by a

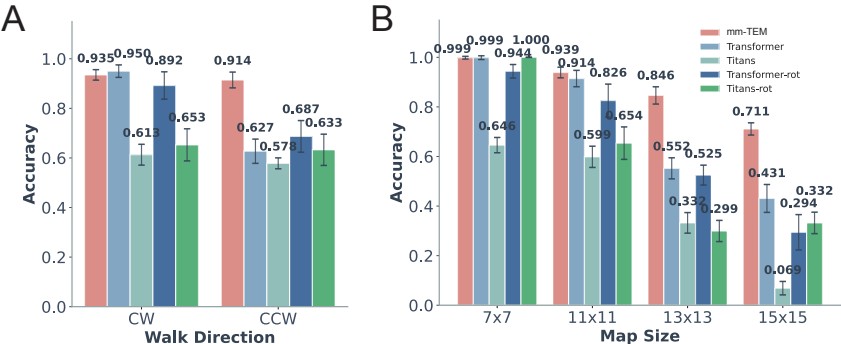

Figure A.4: Comparison of Model Generalization in Circular-grid Environments. All model architectures and task settings are the same as that in Fig. 3 and Fig. A.3.(A) Clockwise vs counterclockwise performance across different architectures. (B) Effect of environment size, ranging from 7×7 to 15×15. All results are averaged over 3 seeds.

wide margin, demonstrating the complementary strengths of short-range memory (Transformer) and long-range, structure-aware memory (mm-TEM). (4) Importantly, mm-TEM and Hippoformer achieve these results with fewer parameters than the Transformer and Titan baselines (see Tab. A.2), indicating that the gains are not due to model size, but architecture.

why PE alone cannot explain the gains. Strong positional encodings (rotary or dynamic PI-based) improve short-range prediction, but they do not endow baseline Transformers and Titans with hippocampus-like computational mechanisms, such as feedback-based error correction or relational memory. Simply inserting a recurrent path integration module as a positional embedding does not yield hippocampus-like abstract structure learning or spatial rule inference, which are essential for multi-step relational reasoning and long-range prediction.

## A.8 Additional ablation and control studies

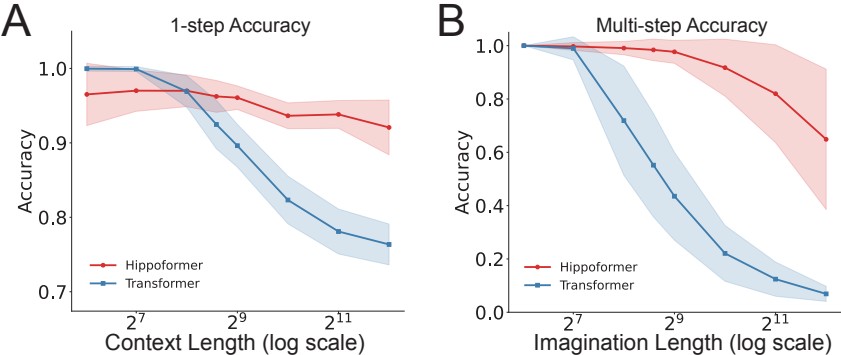

Figure A.5: Hippoformer using Truncated-BPTT vs Transformer using Full-BPTT. Hippoformer has the same architecture as that in Fig. 5. Transfomer uses the same architecture as that in Fig. 3. All models are trained in 2D grid prediction tasks (training length = 512). BPTT in Hippoformer is truncated at 256 steps, and Full BPTT is adopted in Transformer. (A) 1-step prediction accuracy vs context length. (B) Multi-step imagination accuracy vs imagination length. All results are averaged over 3 seeds.

As shown in Fig. A.5, to examine whether full BPTT is necessary, we conducted a new experiment with 512-length sequences. In the task, Hippoformer uses BPTT truncated at 256 steps. Transformer uses full BPTT over all 512 steps. Despite the truncated training, Hippoformer still outperforms the Transformer, demonstrating that full BPTT is not re-

quired for Hippoformer to capture long-range dependencies. In fact, long-range dependency does not rely solely on long unrolled recurrence. It also relies on the architectural prior to learn abstract structural codes.

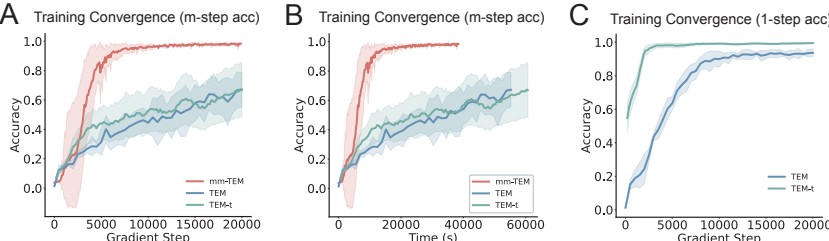

Figure A.6: Training Efficiency and Multi-step Prediction Performance of mm-TEM versus TEM-t Baselines. Task settings and evaluations are the same as that in Fig. 2. (A) Multi-step prediction accuracy during training (as a function of gradient steps). mm-TEM converges substantially faster and achieves markedly higher accuracy than both TEM and TEM-t. (B) Multi-step prediction accuracy plotted against wall-clock training time, showing mm-TEM's superior time-efficiency. (C) Both TEM and TEM-t achieve high one-step ahead prediction accuracy during training. One-step prediction was evaluated using a context sequence length of 128, consistent with the original TEM and TEM-t studies (Whittington et al., 2021; 2020). Notably, TEM-t converges faster than TEM. All results are averaged over 3 seeds.

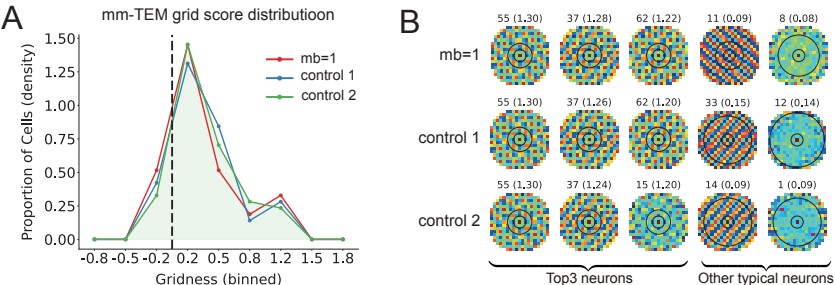

Figure A.7: Ablations Separating Training-time vs. Test-time Effects of (mb). All mm-TEM models are trained with $mb = 1$. In Control 1, the model is tested with $mb = 8$ for both sensory error correction and mm-MLP memory updates. In Control 2, the model is tested with $mb = 8$ only for mm-MLP memory updates, while sensory error correction remains at $mb = 1$. (A) Histogram of grid-cell gridness scores. (B) Autocorrelation maps of grid representations under the different conditions.

According to the above detail feedback mechanism, increasing $mb$ reduces both the frequency of memory updates and the frequency of error-correction feedback. To disentangle these two factors, we performed the following control studies: (1) in Control 1, mm-TEM is trained with $mb = 1$, but tested with $mb = 8$ for both error correction and mm-MLP memory updates; (2) in Control 2, mm-TEM is trained with $mb = 1$, but tested with $mb = 8$ for mm-MLP memory updates while keeping error correction at $mb = 1$. In both cases, grid scale remains identical to the $mb = 1$ model, as illustrated in Fig. A.7. This outcome is expected. In mm-TEM, the grid scales are determined solely by how the path-integration network maps actions into grid representations and by the specific grid–sensory pairings stored in relational memory. The value of $mb$ does not influence either the mapping of path-integration network or the specific grid–sensory pairings, and therefore does not alter the resulting grid scales.

Spearman correlation between grid score and model accuracy is shown in . Note the non-linear correlation significance is more significant than linear.

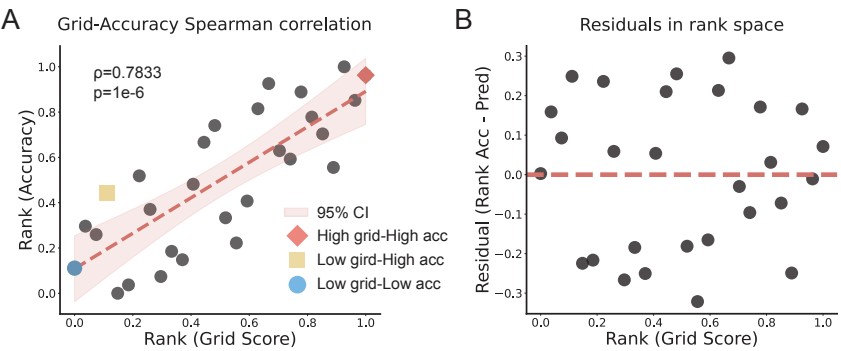

Figure A.8: Spearman correlation between grid score and model accuracy.(A) In the rank space, multi-step prediction accuracy (imagination length = 512) positively correlates with grid score in mm-TEM path integration networks (r = 0.7833, p = 1e-6), indicating that stronger grid-like regularity supports better generalization. Related to Fig. 4A. (B) The residuals.

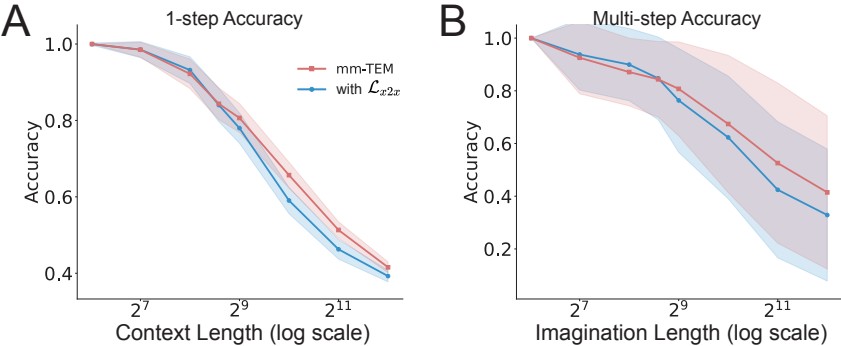

Figure A.9: Control Study On the Auxiliary Relational Loss $\mathcal{L}_{x2x}$. "With x2x" denotes the variant that adds $\mathcal{L}_{x2x}$ to the auxiliary relational loss $\mathcal{L}_{rel}$ in mm-TEM, whereas "mm-TEM" refers to the original model using $\mathcal{L}_{rel}$ without $\mathcal{L}_{x2x}$. (A) and (B) report 1-step and multi-step prediction accuracy, respectively, following the same training and evaluation settings as Fig. 3.

## A.9   OTHER SUPPLEMENTARY RESULTS

Other supplementary results corresponding to the Results in the main text are shown as follows.

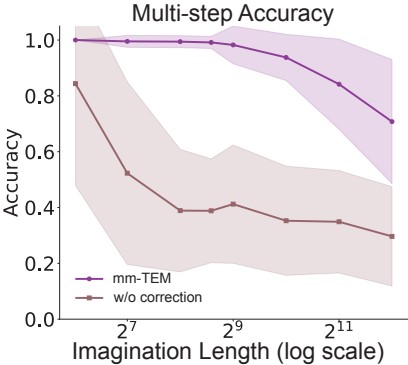

Figure A.10: Both mm-TEM variants - with and without error correction - are trained on sequences of length 256, with $mb = 8$. The models are evaluated using multi-step prediction, and all reported results are averaged over three random seeds.

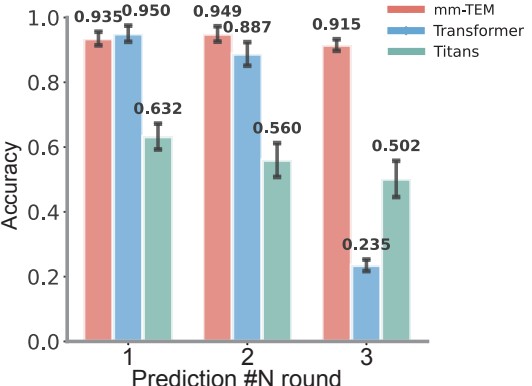

Figure A.11: Generalization Capacity of mm-TEM in Long-Context Conditions. we evaluates the generalization capacity of mm-TEM on the $n$-round imagination task in a circular grid environment, the same as that in Fig. 3D. As the agent explores the environment in a clockwise manner and accumulates more rounds of sensory experience, mm-TEM achieves robust performance, maintaining around 90% accuracy in multi-step imagination. In contrast, both Transformers and Titans exhibit a sharp performance drop as the context input length increases, highlighting the superior generalization ability of mm-TEM under long-context conditions.

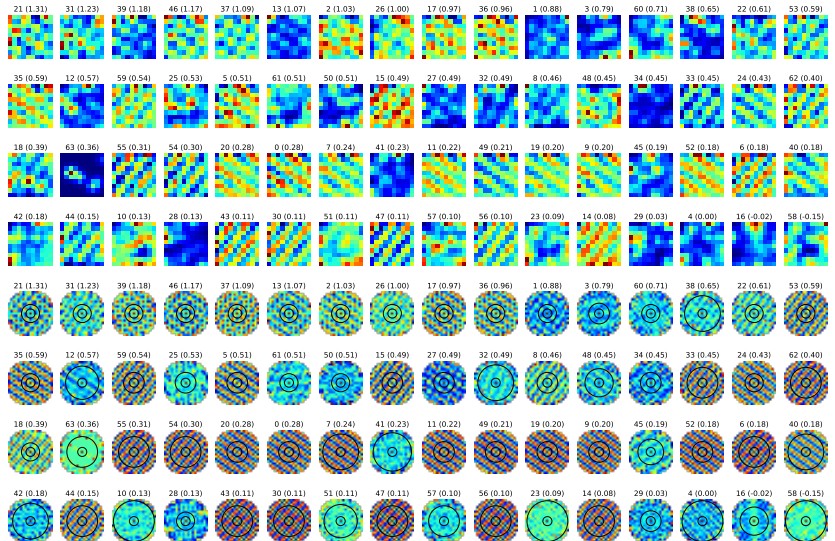

Figure A.12: Grid patterns from mm-TEM with parameter $mb$=1. The upper 4 rows show the ratemaps for each neuron, arranged by grid scores. The lower 4 rows show the auto correlation of ratemaps. Related to Fig. 2C.

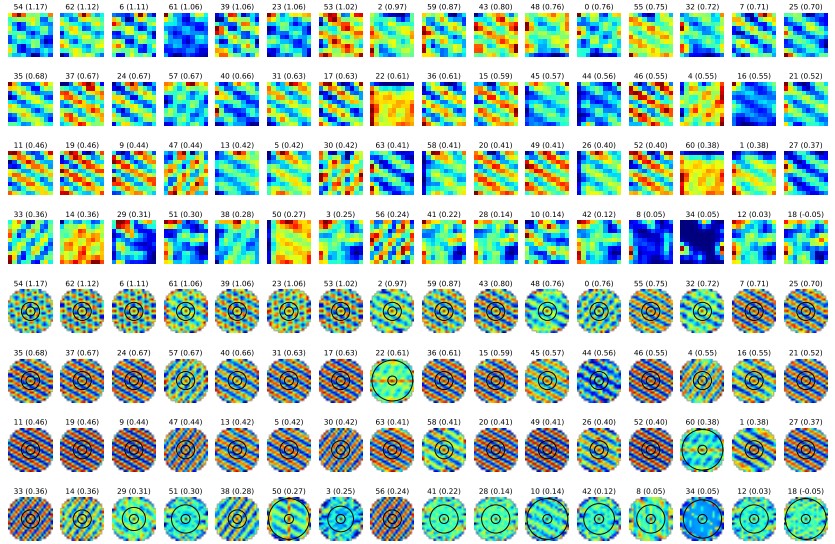

Figure A.13: Grid patterns from mm-TEM with parameter $mb$=4. Related to Fig. 2C.

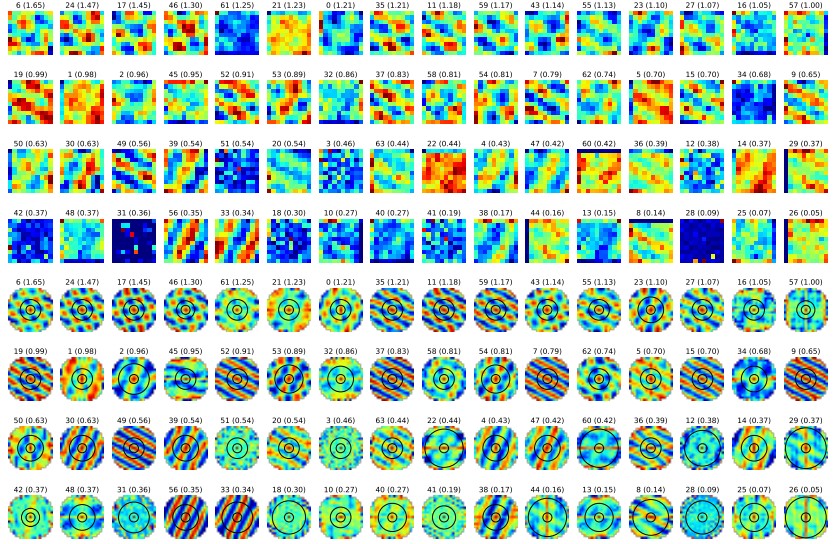

Figure A.14: Grid patterns from mm-TEM with parameter $mb$=8. Related to Fig. 2C.

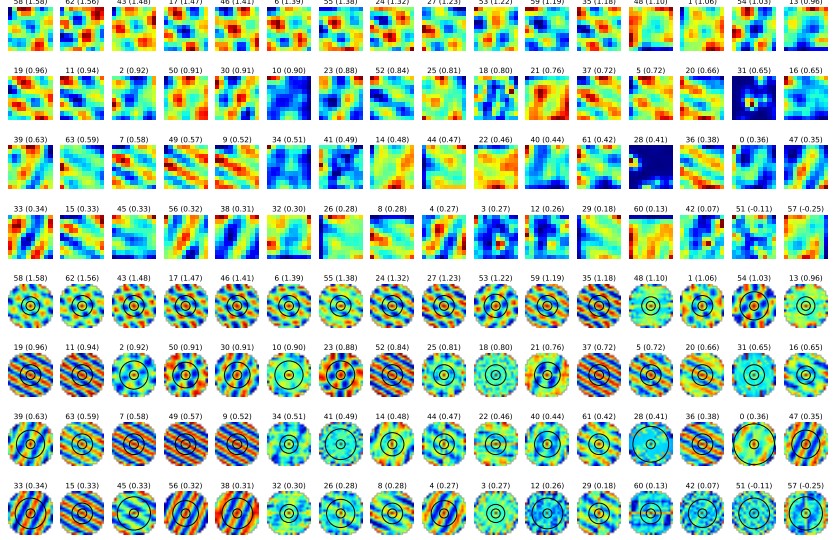

Figure A.15: Grid patterns in representative mm-TEM model whose gridness and accuracy are high. Related to Fig. 4B.

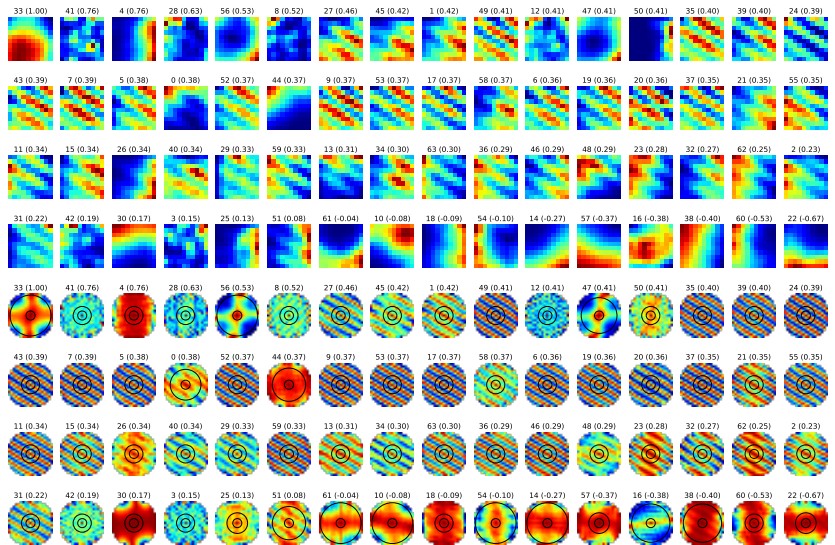

Figure A.16: Grid patterns in representative mm-TEM model whose gridness is low and accuracy is high. Related to Fig. 4B.

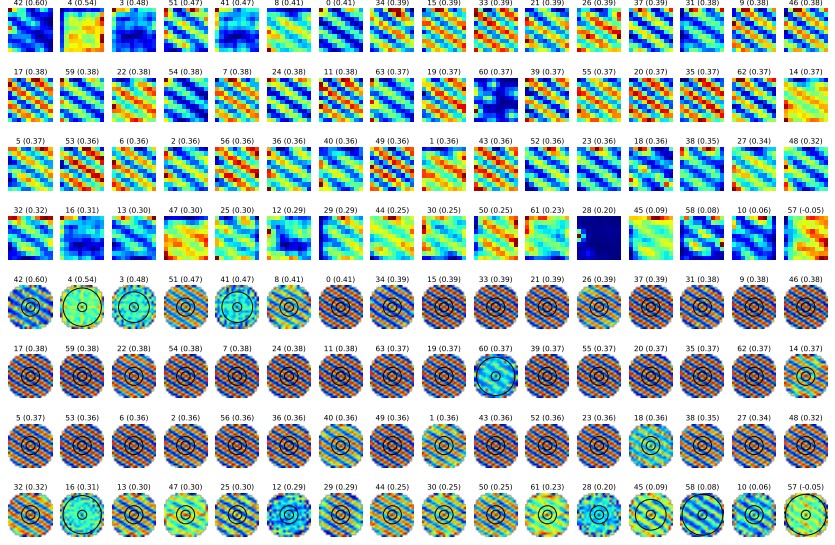

Figure A.17: Grid patterns in representative mm-TEM model whose gridness and accuracy are low. Related to Fig. 4B.

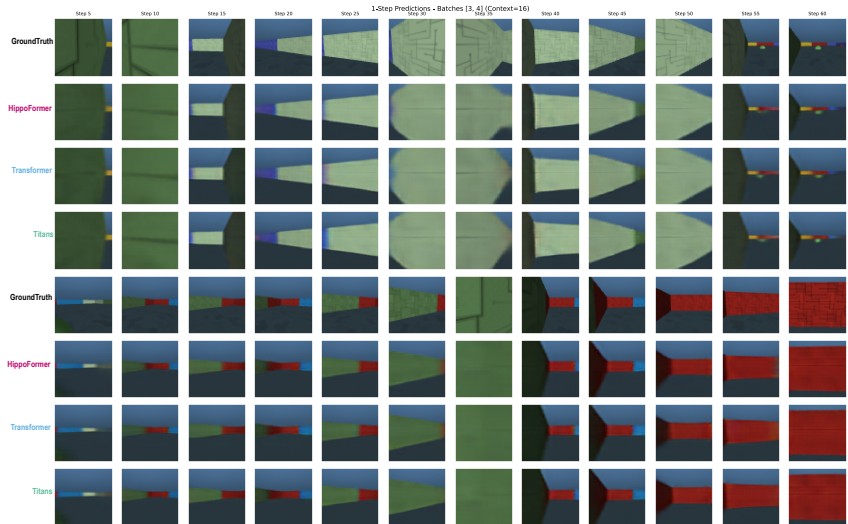

Figure A.18: Visualization of example trajectories of one-step prediction in two 3D environments from different models, with snapshots shown every 5 steps. Related to Fig. 6C.

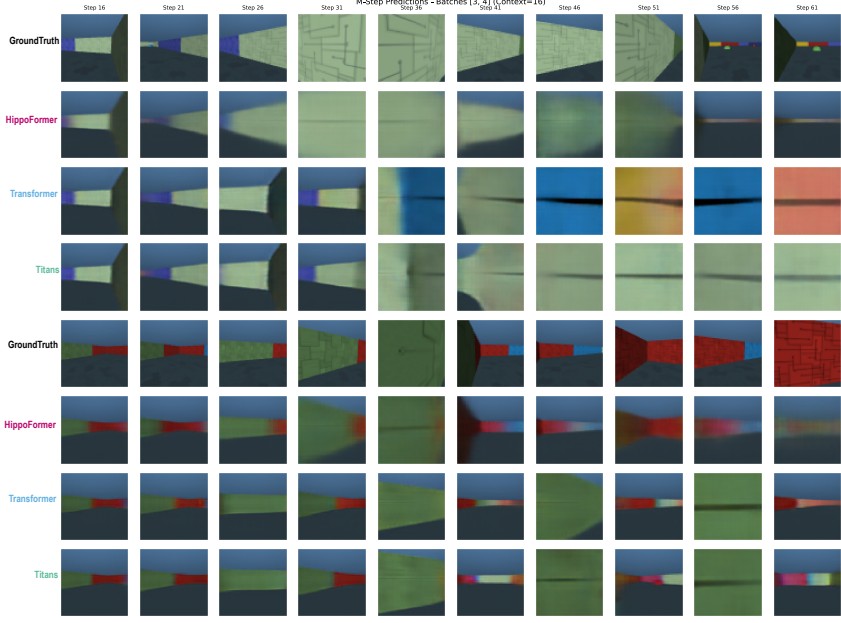

Figure A.19: Visualization of example trajectories of multi-step prediction in two 3D environments from different models, with snapshots shown every 5 steps. Related to Fig. 6C.

