# OpenReview forum: "Hippoformer: Integrating Hippocampus-inspired Spatial Memory with Transformers"
_ICLR.cc/2026/Conference — ICLR 2026 Poster_

### Official Review · Reviewer_3ncY · 2025-10-23

**Soundness:** 2
**Presentation:** 2
**Contribution:** 2
**Rating:** 2
**Confidence:** 5

**Summary:**

This paper proposes mm-TEM, a variant of the Tolman-Eichenbaum Machine using
meta-MLP memory, and Hippoformer, which integrates mm-TEM with Transformers. The
authors evaluate these models on 2D grid prediction and 3D environment tasks,
claiming superior generalization and discovering that memory update frequency
affects grid-like representation scales.

**Strengths:**

* Implementation efficiency: mm-TEM trains faster than original TEM (in gradient * steps), making hippocampal-inspired architectures more practical - but see note below regarding "efficiency"
* Integration with Transformers: The Hippoformer architecture combining structured spatial memory with Transformer's working memory is conceptually interesting.
* Extensive empirical evaluation: The paper includes multiple experimental settings (varying context lengths, environment sizes, circular grids, 2D and 3D tasks).
* Clear presentation: The paper is generally well-written with good figures illustrating the architecture and results.
* Emergence of grid-like representations: The spontaneous development of grid patterns provides interesting connections to neuroscience.

**Weaknesses:**

## Missing Foundational Literature on Hippocampal-Entorhinal Memory Systems

The introduction would be strengthened by acknowledging the broader foundational
literature on hippocampal-entorhinal contributions to episodic and relational
memory beyond spatial navigation. Several core ideas presented as novel to the
TEM framework—namely, factorized MEC-LEC streams, hippocampal binding of
structural and sensory codes, and generalization across relational spaces—have
substantial precedent in earlier empirical and computational work. The 'binding
of items and context' (BIC) model (Eichenbaum et al., 2007) directly addresses
how the hippocampus binds contextual and item information. Lesion and recording
studies have established MEC vs LEC functional dissociations (Hargreaves et al.,
2005; Knierim et al., 2014). The Complementary Learning Systems framework
(McClelland, McNaughton, & O'Reilly, 1995) proposed separate systems for rapid
relational binding and structured representations decades earlier. Computational
models have explicitly implemented factorized spatial and sensory
representations: Hasselmo et al. (2002) modeled grid cells for memory with
structural/contextual codes; Franzius, Sprekeler & Wiskott (2007) demonstrated
grid/place code emergence with associative mapping to sensory inputs; Bush et
al. (2015) explicitly separated grid-cell path integration from memory
association in recurrent networks. I recommend that the authors broaden their
introductory discussion to acknowledge this literature and clarify precisely
what mm-TEM contributes beyond these established frameworks rather than
appearing to introduce factorized hippocampal memory de novo.


## Terminology: "TEM Theory" Should Be "TEM Model/Framework"

Throughout the manuscript, the authors refer to "TEM theory" (e.g., lines 54,
134). I suggest replacing this with "the TEM model" or "the TEM framework." The
Tolman-Eichenbaum Machine (Whittington et al., 2020) is a computational
instantiation of existing theoretical ideas about cognitive mapping and
relational memory, rather than a full-fledged theory in itself. The underlying
theoretical principles, in particular that the hippocampal-entorhinal system
implements factorized memory for flexible generalization, predate TEM by decades
(see comment above). Calling TEM a "theory" risks overstating its conceptual
novelty and may mislead readers about its epistemic status. The contribution of
Whittington et al. (2020) was to provide an elegant computational implementation
of these principles, not to propose the theoretical framework itself. This
distinction matters for accurately situating the current work in the scientific
literature.


## Missing Related Work on Factorized Memory Architecturs
The paper would benefit from acknowledging a broader range of computational
models that have explored factorized memory architectures combining structural
and sensory representations. Beyond the TEM lineage cited, several prior works
share the core principle of separating spatial/structural codes from
sensory/content codes for flexible memory retrieval. Hasselmo et al. (2002)
modeled how entorhinal cortex provides structural/contextual codes while the
hippocampus performs binding operations. The Complementary Learning Systems
framework (McClelland, McNaughton, & O'Reilly, 1995), while not specifically
EC-HC focused, proposed separate systems for rapid relational binding versus
structured long-term representations—a conceptual predecessor to factorized
memory architectures. Franzius, Sprekeler & Wiskott (2007) demonstrated
grid/place code emergence with associative mapping to sensory inputs, providing
an early computational analogue of separating structural representations from
sensory mappings. Bush et al. (2015) explicitly implemented computational
separation of grid-cell-like path integration from memory association in
recurrent networks. Stachenfeld et al. (2017) modeled the hippocampus as a
predictive map using successor representations, integrating spatial structure
with value predictions. Waniek (2020) proposed Transition Scale-Spaces that
integrate structural and sensory information in multi-scale frameworks. Even
work on sequence memory in recurrent networks (e.g., Rajan, Harvey, & Tank,
2016) employs separate state/structure representations with readout layers,
loosely corresponding to mm-TEM's structural versus sensory binding.
Acknowledging these precedents would provide better context for understanding
how mm-TEM's specific implementation choices (meta-MLP memory, auxiliary losses,
Transformer integration) extend versus replicate prior ideas, and would help
readers assess the true novelty of the contribution.


## Imprecise Terminology: "Novelty or Surprisal"
The manuscript refers to "novelty or surprisal" in describing the fast-weight
update mechanism (lines 171-172). In formal information theory, these are
distinct concepts: surprisal is a probabilistic quantity (−log P(x)), whereas
novelty is relative to memory or prior experience (Palm, 2012). What the model
actually computes is the gradient of reconstruction loss, ∇_Θ L(k_t, v_t), which
measures prediction error -- the mismatch between predicted and actual values. I
recommend clarifying which quantity is actually computed and adjusting the
terminology to reflect the implementation accurately. The most precise term
would be "prediction error" or "reconstruction error" rather than the ambiguous
"novelty or surprisal." This precision matters for readers seeking to understand
the computational mechanism and for those attempting to reproduce or extend the
work.


## Misleading Description of Hippocampal-Entorhinal Feedback
In Appendix A.2 (lines 630-632), the manuscript states: "visual sensory cues
provide feedback from the HC to correct path integration errors in the MEC."
This description is biologically misleading. While feedback from the hippocampus
can indeed help stabilize MEC representations and correct errors in path
integration (Diehl et al. 2019, Mulas et al. 2016, and many others), the hippocampus does not send raw visual sensory information to
MEC. Visual and other sensory inputs primarily reach MEC via cortical pathways
(particularly through LEC from perirhinal and parahippocampal cortices). A more
accurate description would be that the hippocampus provides relational or
spatial feedback, likely in the form of conjunctive codes that bind spatial and
sensory information, that can help recalibrate MEC structural representations,
while MEC continues to integrate sensory cues from neocortical areas. I
recommend rephrasing this passage to clarify that the feedback represents a
computational abstraction where HC provides spatial/relational corrections
rather than literal sensory signals. This distinction is important for readers
interpreting the model in a neurobiological context and for understanding what
aspects of the architecture are biologically plausible versus computational
conveniences.


## Missing Critical Baseline: Direct Associative Memory Comparison
The paper's central claim is that factorizing memory into structural codes (via
path integration) and sensory codes enables superior generalization compared to
standard architectures. However, the experimental evaluation lacks a critical
ablation: a direct associative memory baseline that learns state-transition
pairs (state_t, action_t) → state_{t+1} using the same meta-MLP architecture,
warm-up procedure, and auxiliary losses, but without structural factorization.
The current baselines (Transformer, Titans) differ from mm-TEM in multiple
confounded ways -- architecture, training procedures, and task formulation --
making it impossible to isolate whether the performance gains stem from the
factorized representation itself or simply from having a meta-trained
associative memory.  The ablations in Fig. 3C only remove auxiliary losses, not
the core architectural choice of factorization. Furthermore, the claim of
"efficiency" lacks rigor: no computational complexity analysis, wall-clock time
comparison, or memory footprint evaluation is provided, only gradient step counts
against the original TEM. The comparison is further confounded by the warm-up
pre-training phase whose application to baselines is unclear. Finally, in small
grid worlds (8×8 to 11×11), a 64-step context likely covers a substantial
fraction of reachable states, meaning the model may largely be performing
within-episode memory retrieval rather than true generalization to novel spatial
configurations. A proper ablation study isolating the contribution of structural
factorization is essential to validate the paper's core hypothesis.

## Insufficient Mechanistic Explanation for Grid Scale vs. Update Frequency Relationship
The paper claims that the memory update frequency hyperparameter mb controls
grid scale through an "effective prediction horizon" mechanism (lines 252-256),
positioning this as a novel insight into grid-scale diversity. However, this
explanation lacks mechanistic rigor and conflates multiple confounded factors.
First, mb affects both training dynamics (how often gradients update the
meta-MLP weights) and inference behavior (how stale the memory becomes), but the
paper does not disentangle which factor drives the grid scale effect. The
observed correlation could simply be an artifact of optimization
dynamics. That is, sparser gradient updates naturally produce temporally smoother,
lower, frequency representations through gradient accumulation, rather than
reflecting a meaningful "prediction horizon." Second, the feedback mechanism
from relational memory to path integration (Appendix A.2) is central to
understanding this effect but is incompletely described: the function f_delta is
not defined, the strength of feedback (α) is not analyzed, and how mb interacts
with this feedback is unclear. Third, no ablation studies isolate the causal
mechanism. For instance, training with mb=1 but testing with mb=8, or analyzing
gradient flow as a function of mb. Without this mechanistic analysis, the claim
that mb reveals insights into biological grid-scale diversity through
"multi-timescale predictions" remains speculative correlation rather than
demonstrated causation. The authors should provide: (1) ablations separating
training-time vs. test-time effects of mb, (2) gradient flow analysis explaining
why different update frequencies produce different spatial frequencies, and (3)
complete mathematical description of the feedback mechanism.

## Missing Citation and Overclaimed Novelty on Grid Scale Mechanisms
The paper claims to reveal "a novel mechanism for grid-scale diversity in
MEC...as a natural consequence of multi-timescale predictions in the brain"
(lines 252-256), based on their observation that the memory update frequency mb
affects grid scale. However, this is not novel -- Waniek (2020, "Transition
Scale-Spaces: A Computational Theory for the Discretized Entorhinal Cortex")
analytically derived that grid scales emerge from different prediction horizons,
with the spatial frequency directly related to the temporal prediction distance.
The current paper essentially re-discovers this relationship empirically without
citing this prior work. Moreover, Waniek's analysis is rigorous in the sense of
an analytical derivation, compared to the current paper's informal "effective
prediction horizon" argument. Related work by Stachenfeld et al. (2017) and
Dordek et al. (2016) also established connections between temporal prediction
scales and spatial grid scales. The authors should: (1) cite this prior
theoretical work, (2) clarify what is actually novel about their contribution
beyond empirical confirmation in a different architecture, and (3) either remove
the novelty claim or demonstrate what their mechanism adds beyond existing
theory.

## Ambiguity in Generalization Claims Across Sections 3.1 and 3.2
The paper creates confusion about what "generalization" means across different
experiments. In Appendix A.1 (lines 600-602), the authors state that for "all 2D
grid prediction tasks," evaluation is confined to previously visited positions
because "predicting observations at unvisited locations is not meaningful" given
that observations are uncorrelated discrete IDs. However, in Section 3.2, which
claims to test "generalization" (line 260), this critical constraint is never
restated, leaving readers to infer that it still applies. This matters because
Section 3.2's "multi-step imagination" is framed as testing the model's ability
to "generalize beyond its training horizon" (line 268), but if evaluation is
confined to previously encountered positions, it's actually testing memory
retrieval over longer sequences, not spatial generalization to novel states. The
paper should explicitly clarify for each experiment: (1) whether evaluation
includes unseen positions, (2) what fraction of test positions were visited
during context, and (3) how "accuracy" is computed when positions are/aren't
previously seen. The distinction is crucial: the 3D experiments (Section 3.4)
acknowledge that "unvisited observations can be inferred from nearby spatial
information" due to continuous visual features, but no such acknowledgment is
made for 2D tasks where this fundamental difference in evaluation regimes should
be highlighted upfront in Section 3.2, not buried in the appendix.


## Unjustified Causal Claims from Correlational Data (Section 3.2, Figure 4)
The paper claims that "the presence of strongly grid-like cells is a key driver
for generalization" (lines 346-347) based solely on a correlation between grid
score and prediction accuracy (r=0.647, Fig. 4A). This is a causal claim
unsupported by the evidence. Correlation does not establish causality -- the
relationship could be due to reverse causation (good generalization enables
better grid formation) or a confounding variable (successful learning produces
both regular representations and good generalization). Notably, the paper's own
data weakens the causal claim: Figure 4B shows models with low grid scores
achieving high accuracy through "alternative but still regular neural
representations," suggesting that regularity (not grid-ness specifically) may be
what matters. To establish causality, the authors should conduct interventional
experiments: (1) inject hand-designed grid representations and test if
performance improves, (2) add regularization to suppress grid formation and test
if performance degrades, or (3) explicitly bias learning toward grid patterns
and compare against controls. As written, the claim that grids "facilitate" or
"drive" generalization is speculative. The authors should either provide causal
evidence or rephrase their claims to accurately reflect the correlational nature
of their findings (e.g., "grid scores correlate with generalization performance,
suggesting a potential relationship").


## Questionable Statistical Analysis in Figure 4A
The correlation analysis in Figure 4A raises several statistical concerns.
First, while reporting r=0.647 (p=0.0002), the authors use linear regression
despite evidence of non-linearity: accuracy appears to show a ceiling effect
(~0.95-1.0) and the relationship exhibits heteroscedasticity (variance in
accuracy is much higher at low grid scores than high). This violates key
assumptions of linear regression, making standard errors, confidence intervals,
and p-values unreliable. Second, with only ~20-25 data points visible, the
analysis is sensitive to outliers and the wide confidence interval suggests
substantial uncertainty. Third, r²≈0.42 means 58% of variance in accuracy
remains unexplained, yet the authors make strong causal claims ("key driver")
from this weak-to-moderate correlation. Fourth, the scatter plot shows several
counterexamples to the claimed relationship: models with grid scores around
0.7-0.9 achieve accuracy >0.9, while models with grid scores ~1.0-1.1 have
accuracy ~0.8. These observations, also noted in the text regarding "low grid
scores still achieve high accuracy" (Fig 4B), directly contradict the linear
relationship implied by the regression line. The authors should: (1) test for
non-linear relationships (threshold models, saturation functions), (2) report
non-parametric correlations (Spearman's ρ) that don't assume linearity, (3) show
residual plots and test assumptions, (4) report prediction intervals to
demonstrate the large uncertainty in predictions, and (5) acknowledge that grid
score alone is a poor predictor of performance. The statistical evidence does
not support the strong causal claims made in the text.

## Undefined Error Metric and Inconsistent Terminology in 3D Experiments (Section 3.4, Table 1)
The 3D environment evaluation suffers from unclear methodology and inconsistent
reporting. Table 1 reports "prediction error in units of 1e-3" but never defines
what error metric is used. Presumably MSE between predicted and ground truth
egocentric images, but this is not stated. Are errors computed in pixel space,
normalized space, or feature space? How are images preprocessed? The text
compounds confusion by referring to "accuracy" (lines 432-435) while the table
shows "error". These are opposite metrics (higher accuracy vs. lower error is
better). The classification of frames as "visible" vs "not visible" is
undefined: what determines if a frame is considered previously seen in
continuous 3D space with egocentric views? The reported standard deviations
raise questions: one-step errors show std=0.00 across 3 seeds for all models
(presumably due to rounding, but this should be clarified), while Hippoformer's
multi-step std (0.04) is 100× smaller than baselines (4-5). Why is Hippoformer so
much more stable? Finally, without baseline comparisons (chance level, naive
predictors) or interpretation of error magnitudes (e.g., "0.001 corresponds to X
per-pixel deviation"), the numbers lack context. The authors should: (1)
explicitly define the error metric and computation procedure, (2) use consistent
terminology (error or accuracy, not both), (3) specify the visible/not-visible
classification criterion, (4) explain the variance patterns across models, and
(5) provide baselines and interpretation to make the magnitudes meaningful.


## Discussion Section Overclaims Novelty and Omits Critical Limitations
The Discussion overclaims novelty and omits acknowledgment of significant
limitations identified throughout the paper. First, the claim that mm-TEM offers
"a new functional perspective on grid diversity" (line 445) ignores Waniek
(2020), who analytically derived that grid scales emerge from prediction
distances - the very relationship mm-TEM rediscovers empirically. Second, the
Related Work section omits foundational hippocampal-entorhinal literature
(Eichenbaum et al., 2007; Hasselmo et al., 2002; Stachenfeld et al., 2017;
Franzius et al., 2007; Bush et al., 2015) that established factorized memory
architectures and prediction-timescale-to-spatial-scale relationships decades
before TEM. Third, while the authors acknowledge limited integration and
single-layer design, they fail to address major methodological limitations: (1)
2D evaluation confined to previously visited positions (Appendix A.1), meaning
"generalization" is actually memory retrieval, not spatial inference to novel
states; (2) small environments (8×8 to 11×11) where 64-step context covers
substantial state space; (3) no ablation isolating the contribution of
factorization versus direct associative memory; (4) causal claims about
grid-generalization relationship based solely on correlation (r=0.647); (5)
undefined error metrics and inconsistent terminology in 3D experiments. Fourth,
efficiency and scalability claims lack rigor: no computational complexity
analysis, wall-clock time comparisons, or large-scale demonstrations are
provided. The Discussion should thus (1) cite prior work and clarify what is
novel about the grid-scale finding beyond empirical confirmation, (2)
acknowledge foundational neuroscience literature on factorized memory, (3)
explicitly discuss the limitation that 2D "generalization" is constrained to
previously visited positions (or clarify the text) (4) acknowledge that
grid-generalization causality remains unproven, (5) provide concrete criteria
for when mm-TEM/Hippoformer would be preferred over standard architectures, and
(6) temper claims about efficiency and scalability until rigorous evidence is
provided.

**Questions:**

1. Can you provide a direct ablation comparing mm-TEM against a flat associative memory (no factorization) with identical meta-MLP architecture, warm-up procedure, and auxiliary losses?
2. Can you clarify the evaluation protocol for 2D tasks in Section 3.2? What percentage of positions in the "imagination" phase were previously visited during context?
3. For the mb parameter effect on grid scales: Can you ablate training vs. test-time effects? (Train with mb=1, test with mb=8 and vice versa?)
4. For Figure 4A: Can you provide non-parametric correlation measures (Spearman's ρ), test for non-linear relationships, and show residual plots?
5. For Table 1: What specific error metric is used? How are "visible" vs. "not visible" frames classified in continuous 3D space?
6. Can you provide wall-clock time comparisons and computational complexity analysis to support efficiency claims?
7. How does performance scale to truly large environments where 64-step context covers <10% of reachable states?

---

> ### Author Response · Authors · 2025-11-22
> **Response to Reviewer 3ncY. Part 1**
>
> We are grateful to the reviewer for the extensive and detailed feedback. The detailed points raised—spanning methodology, interpretation, prior literature, and presentation—have been valuable in helping us refine the framing and strengthen both the empirical and conceptual components of the work. We address each issue carefully in the responses that follow.
>
> ### On Weaknesses:
>
> **1. Missing Foundational Literature on Hippocampal-Entorhinal Memory Systems**
>
> We thank the reviewer for the careful and insightful comments. We fully agree that our work is grounded on a substantial body of literature on hippocampal–entorhinal contributions to episodic and relational memory. Due to space limitations, our initial version of manuscript  does not include several key empirical and computational frameworks. We appreciate the reviewer’s detailed pointers to these important works. We now recognize that this was inappropriate, and we have made extensive additions and revisions to properly address and incorporate these references.
>
> In response to Weakness 1, we have revised the Introduction to explicitly cite the BIC model, MEC–LEC dissociation studies and prior computational models using factorized spatial and sensory representations. For completeness, we reproduce the revised paragraph below (also updated in the manuscript):
>
> >The hippocampal-entorhinal (HC–EC) system is central to spatial and episodic memory (Buzsáki & Moser, 2013; Eichenbaum et al., 2007; Eichenbaum, 2017; Whittington et al.,2022). Neurophysiological studies have identified two key computational principles: (1) the functional dissociation between MEC and LEC pathways for structural and (2) sensory information (Hargreaves et al., 2005; Knierim et al., 2014), and the hippocampus’s role in binding them (Eichenbaum et al., 2007). These insights have inspired a range of computational models with factorized structural and sensory representations (Hasselmo et al., 2002; Franzius et al., 2007; Bush et al., 2015), where MEC provides path-integration–based structural codes and the hippocampus binds them with LEC-derived sensory inputs. Building on this, several learnable HC–EC–inspired models have emerged, including CSCG (George et al., 2021; Raju et al., 2024), the Tolman–Eichenbaum Machine (TEM) (Whittington et al., 2020), and Vector-HaSH (Chandra et al., 2025). Among these, TEM offers a unified computational framework that learns abstract structure in an unsupervised manner and generalizes this structure to novel environments (Fig. 1A-B), making it a promising basis for hippocampus-like memory systems. Despite these advances, most existing HC–EC models focus mainly on simplified synthetic settings (Whittington et al., 2020; Raju et al., 2024; Zou et al., 2024; Chandra et al., 2025). Bridging these biologically grounded mechanisms with modern deep learning architectures—and scaling them to complex, real-world tasks—remains a major open challenge.
>
> **2. Terminology: "TEM Theory" Should Be "TEM Model/Framework"**
>
> We thank the reviewer for this important clarification. We agree that the TEM is a computational model/framework that instantiates long-standing theoretical principles rather than constituting a new theory in itself. We have updated all 'TEM theory' to 'TEM model / framework' accordingly in the revised manuscript. And this correction indeed increases the rigor of our paper.
>
> **3. Missing Related Work on Factorized Memory Architectures**
>
> We appreciate the reviewer’s thorough list. We have significantly expanded the related works to explicitly discuss prior factorized memory models, both in the Introduction and Method (see our reply to **Weakness 1** and the revised manuscript).
>
> **4. Imprecise Terminology: "Novelty or Surprisal"**
>
> We appreciate the clarification. The mechanism indeed computes a prediction error. We have revised "**novelty or surprisal**" to "**prediction error**" in the revised manuscript.
>
> **5. Misleading Description of Hippocampal-Entorhinal Feedback**
>
> Thank you for pointing out this inaccuracy. We have rephrased the passage to clarify that the hippocampus provides relational/spatial feedback—rather than sensory signals—to help recalibrate MEC representations, to avoid misunderstanding.
>
> The passage "visual sensory cues provide feedback from the HC to correct path integration errors in the MEC" has been modified as:
> >In the HC–EC system, the hippocampus provides relational and spatial feedback - likely in the form of conjunctive codes that bind spatial and sensory information - to help recalibrate MEC structural representations (Diehl et al., 2018; Mulas et al., 2016).

---

> ### Author Response · Authors · 2025-11-22
> **Response to Reviewer 3ncY. Part 2**
>
> **6. Missing Critical Baseline: Direct Associative Memory Comparison**
>
> Thank you for the insightful comments. We would like to re-emphasize that in mm-TEM, both the relational memory (implemented by the meta-MLP with auxiliary relational loss) and the factorized structure and content codes are crucial, as highlighted in the Introduction. In unsupervised learning, these two mechanisms interact synergistically, thereby facilitating the formation of abstract spatial structure within the path integration network (verified by the emergence of the grid pattern).
>
> We have revised the manuscript and added new experiments to address the proposed concerns.
>
> **Direct Associative Memory Baseline**:
>
> Titans, based on the same meta-MLP fast-weight memory, already served as a direct associate memory baseline that explicitly learn the state-transition mapping $(s_t, a_t) \rightarrow s_{t+1}$. Input tokens are formed by concatenating the embedding vector of $a_t$ and $s_t$ , then adding positional encoding (details clarified in the revised Method section). Despite using a larger network (30.62 MB in Titans vs. 9.11 MB in mm-TEM), Titans performs substantially worse across one-step and multi-step prediction (**Fig. 3A,B,E,F**), indicating that a meta-trained associative memory alone is insufficient for 2D grid tasks .
>
> **Why Warm-Up and Auxiliary Losses Are Not Applied to Baselines**:
>
> The warm-up procedure and auxiliary relational loss are designed specifically to encourage meta-MLP to form conjunctive relational memories. Applying them to the direct transition model $(s_t,a_t) \rightarrow s_{t+1}$ is inappropriate, because the same sensory state can be reached with different actions from distinct contexts, creating conflicting bindings. For instance, the exactly same $s$ can have four bindings, $(s,a_{east})$,$(s,a_{west})$,$(s,a_{north})$ and $(s,a_{south})$.
>
> **Additional Ablations on Positional Information**:
>
> In mm-TEM, path integration network can be seen to act as a dynamic position embedding. To strengthen the associative memory baselines, we added new experiments replacing Titans’ positional information with:
>
> (1) **Sinusoidal Embedding:** A static, absolute positional embedding.
>
> (2) **Rotary Embedding (RoPE):** A relative positional embedding method.
>
> (3) **Dynamic Embedding:** A positional embedding obtained via the integration of action, using the same recurrent network as mm-TEM but without the error correction mechanism.
>
> Results can be seen in **Fig. A.3**. None closed the performance gap with mm-TEM, suggesting the insufficiency of associative memory alone. Notably, the dynamic-embedding variant of Titans, which lacks error correction and thus the necessary interaction between the factorized structure and relational memory, still fails, supporting the necessity of interaction between structure factorization and relational memory.
>
> **Efficiency Analysis**:
>
> To ensure a better comparison, we conducted an additional experiment using TEM-t, a more efficient variant of TEM. As demonstrated by the wall-clock training time **in Fig. A.6A-B of the revised manuscript**, mm-TEM achieves better generalization performance while simultaneously maintaining a faster convergence speed.
>
> **Clarifying Generalization**:
>
>  Our 2D evaluation follows TEM and TEM-t: accuracy is computed only on _visited_ states, because unvisited states are inherently unpredictable in the 2D disctrete grid state setting. But many transitions remain unseen. Even if a state A has been visited, the _transition_ leading from a different state B to A may never have occurred. The model must infer what sensory observation to expect when taking an action from B that leads to A. Successful prediction therefore requires understanding the underlying spatial structure and the action-dependent transition rules.
>
> As shown in Fig. 3D–F, although all nodes are visited in a clockwise traversal, both Transformers and Titans fail on unseen transitions(e.g., counter-clockwise), whereas mm-TEM generalizes correctly by leveraging the learned structural abstraction rather than within-episode memorization alone.
>
> **Summary**:
>
>  In the above reply, we (1) clarified that existing baselines are direct associative memories, (2) added positional-encoding ablations, (3) explained why warm-up/auxiliary losses cannot be applied to direct-transition models, (4) added  efficiency comparisons, and (5) clarified the structural generalization in detail. Together, these results consistently indicate that mm-TEM’s advantage arises from the interaction between structural–sensory factorization and relational memory, rather than associate memory alone.

---

> ### Author Response · Authors · 2025-11-22
> **Response to Reviewer 3ncY. Part 3**
>
> **7. Insufficient Mechanistic Explanation for Grid Scale vs. Update Frequency Relationship**
>
> We thank the reviewer for this important point. In the revision, we have (1) clarified the feedback mechanism in **Appendix A.2**, (2) added control experiments that strengthen our claim and (3) provided an intuitive mechanistic explanation for the relationship between grid scale and update frequency in the discussion. The details are summarized below.
>
> In the revision, we (1) provide a complete description of the feedback mechanism, (2) add ablations disentangling training- vs. test-time effects of mb, and (3) clarify the mechanistic intuition behind the grid-scale relationship.
>
> **Clarifying the feedback mechanism**:
>
> For "f_delta is not defined": the function $f_{delta}$ has been defined in the original manuscript as "two-layer MLP that predict the variance of the integrated structural code" in Sec.3.1 of the main text.
>
> For the reviewer’s concern **“how (mb) interacts with this feedback is unclear”**: In mm-TEM, the meta-MLP memory is updated once every **$mb$** time steps.  At each time $t$, the model aggregates the recent pairs $(g_{gen,t},s_t)$
> collected since the last update. Given an input sequence  $[\dots,(g_{gen,t},s_t),(g_{gen,t+1},s_{t+1}),\dots,(g_{gen,t+mb},s_{t+mb}),\dots]$, if the previous memory update occurs at time $t$, the next one occurs at $t+mb$.
>
> At time $t+mb$, the path integration network is first corrected using $(g_{gen,t+mb},s_{t+mb})$. This pair is sent to the relational memory (meta-MLP), which retrieves a structural code $\hat{g}\_{gen,t+mb}$. . The retrieved code is then combined with the previous structural estimate $g_{gen,t}$ to produce an updated, corrected structural code $g_{inf,t+mb}$. After this correction step, the meta-MLP memory is then updated, using the batch $[(g_{gen,t},s_t),\dots,(g_{gen,t+mb-1},s_{t+mb-1}),(g_{inf,t+mb},s_{t+mb})]$.
>
> Importantly, throughout the interval $t \rightarrow t+mb$, the network has no access to these "future" $(g,s)$ pairs. All predictions must rely solely on information available at and before time $t$. Consequently, the model must implicitly predict 1 to $mb$ steps ahead, and a larger $mb$ increases the effective prediction horizon.
>
>
> **Ablations separating training-time vs. test-time effects of $mb$**:
>
> According to the above detail feedback mechanism, increasing $mb$ reduces both the frequency of memory updates and the frequency of error-correction feedback. To disentangle these two factors, we performed the following control studies:
>
> 1. **Control 1:** train mm-TEM with $mb=1$, but test with $mb=8$ for both error correction and meta-MLP memory updates.
>
> 2. **Control 2:** train mm-TEM with $mb=1$, but test with $mb=8$ for meta-MLP memory updates while keeping error correction at $mb=1$.
>
> In both cases, grid scale remains identical to the $mb=1$ model, as illustrated **in Fig. A.7 of the revised manuscript**.
> This outcome is expected. In mm-TEM, the grid scales are determined solely by how the path-integration network maps actions into grid representations and by the specific grid–sensory pairings stored in relational memory. The value of $mb$ does not influence either the mapping of path-integration network or the specific grid–sensory pairings, and therefore does not alter the resulting grid scales.
>
> **Why a Large Prediction Horizon Encourages Large Grid Scales**:
>
> A possible intuitive explanation from optimization view is: In mm-TEM, the relational memory stores historical $(g, s)$ pairs. Predictions are produced by using the structural code $g$ to query the relational memory and retrieve the associated sensory code $s$.
>
> For **small grid scales**, the structural code is highly precise, but grid codes of neighboring locations are **weakly correlated**. This makes the representation more sensitive to accumulated errors from path integration.
>
> For **large grid scales**, the grid codes are less spatially precise, yet **strongly correlated** across nearby positions. Each code carries richer contextual information about its surroundings and is more robust to accumulated integration error.
>
> Consequently, when the prediction horizon is short (e.g., ($mb = 1$)), accumulated path integration error remains small, so a highly precise spatial code is advantageous. However, when the prediction horizon is long (e.g., $mb = 8$), accumulated error grows because sensory feedback for error correction is sparse (only every $mb$ steps). Under these conditions, a more correlated and robust representation becomes beneficial, making **larger-scale grid codes** preferable during optimization.

---

> ### Author Response · Authors · 2025-11-22
> **Response to Reviewer 3ncY. Part 4**
>
> **8. Missing Citation and Over claimed Novelty on Grid Scale Mechanisms**
>
> We thank the reviewer for highlighting the important works of Waniek (2020), Stachenfeld et al. (2017), and Dordek et al. (2016), all of which link temporal prediction scales to grid spacing. We have cited and acknowledged these works, and have revised our novelty claim accordingly to make it more clear in the results and discussion.
>
> That said, our contribution differs from these works in two concrete ways:
>
> First, prior models do not learn grid patterns with different scales in a concrete HC-MEC model end-to-end: grid scales are analytically derived (Waniek, 2020) or arise as basis functions of predefined multiscale place codes (Dordek et al., 2016; Stachenfeld et al., 2017). In contrast, mm-TEM produces multiple grid scales emergently through self-supervised learning, simply by adjusting the memory-update frequency, without imposing any multiscale place structure. This end-to-end emergence is important because it shows that diverse grid scales can arise naturally from top-down optimization in a biologically inspired system, enabling prior theoretical ideas about multiscale grid representations to be tested and extended in more complex and realistic task settings.
>
> Second, mm-TEM offers a new perspective on the biophysical mechanisms underlying diverse grid scales in the brain. Unlike prior approaches that learn to reconstruct predefined multiscale place representations (Dordek et al., 2016; Stachenfeld et al., 2017), mm-TEM suggests that memory-update frequency itself may play a critical role by implicitly implementing a multiscale prediction horizon. Biologically, variation in memory-update frequency may be able to related to known gradients along the ventral–dorsal hippocampal axis, such as oscillation-frequency gradients (Abhinav Goyal et al., 2020) or receptor-expression gradients (Strange et al., 2014).
>
> Finally, our proposed mechanism is also highly recognized by reviewer 8yz6.
>
>
> **9. Ambiguity in Generalization Claims Across Sections 3.1 and 3.2**
>
> Thanks for the reviewer’s comments. We clarify this point below.
>
> (1) Regarding **“…‘generalization’ (line 260), this critical constraint is never restated …”**, our original manuscript already states this constraint explicitly for the 2D grid task: “Because observations at unseen locations within this discrete environment are unpredictable, both training and evaluation are confined to positions previously encountered within each sequence.” in **Sec.3.1**.
>
> (2) Regarding the comment **“...for testing memory retrieval over longer sequences, not spatial generalization to novel states...”**, our evaluation is in fact non-trivial. The model is _not_ required to generalize to entirely novel states, but rather to **novel transitions (edges) between states**.
>
> This type of evaluation is also used in TEM and TEM-t. Such generalization demands that the network capture the underlying spatial structure and relational rules. Pure long-term memorization alone cannot succeed on these tasks, as evidenced by the performance of Transformers and Titans (Fig. 3A–B).
>
> Moreover, as discussed in the introduction and related work, current deep architectures lack a hippocampus-like spatial memory system and therefore cannot organize long-term memory through abstract spatial structure—an problem clearly articulated in Fei-Fei Li’s work(Jihan Yang et al.,2025). In this sense, organizing long sequences in a spatially structured manner is far from trivial, and may help improve spatial reasoning capabilities in existing architectures.
>
> (3)  Regarding **"what fraction of test positions were visited during context"**, as in 2D grid tasks, generalization occurs over **unseen transitions (edges)** rather than unseen states. Although a long context may cover many nodes, the number of possible transitions between these nodes grows rapidly with sequence length. Therefore, the model cannot rely on simple memorization; it must infer **long-range spatial relations**—that is, the global spatial structure and connectivity implied by local sequential experience—so that it can predict transitions between states that were not directly observed during context.
>
> Additionally, we conducted a statistic analysis of trajectory coverage on a 11×11 rectangular environment using 3200 trajectory samples. The results are shown below.
>
> |Seq Len | State Coverage (%) | Edge Coverage (%) |
> |--------|--------------------|-------------------|
> | 64     | 24.76 ± 4.43       | 9.16 ± 0.82       |
> | 128    | 43.00 ± 6.50       | 17.55 ± 1.35      |
> | 256    | 66.66 ± 7.32       | 31.93 ± 2.13      |
>
> In short, in an 11×11 environment, a randomly sampled 64-step trajectory covers ~25% states and only ~9% edges.
>
> (continued in Part5)

---

> ### Author Response · Authors · 2025-11-22
> **Response to Reviewer 3ncY. Part 5**
>
> (4) Regarding **“how ‘accuracy’ is computed when positions are/aren’t previously seen”**:
>
> Thanks for pointing this out. We have clarified and revised the accuracy computation method in the updated manuscript：
>
> For one-step prediction (Fig. 3A), given a context sequence of length $T_1$ with state-action inputs $[(s_1, a_1), (s_2, a_2), \ldots, (s_{T_1}, a_{T_1})]$, the model generates next-state predictions $\hat{S} =[\hat{s}\_2, \hat{s}\_3, \ldots, \hat{s}\_{T_1}]$. A predicted state is counted as valid only when its corresponding ground-truth was visited before $t$.
>
> For multi-step imagination (Fig. 3B), after receiving a context of length $T_1$, the network performs $T_2$-step imagination. During imagination, an action sequence $[a_{T_1}, \ldots, a_{T_1+T_2}]$ is provided, and the network produces predicted states $\hat{S} = [\hat{s}\_{T_1+1}, \hat{s}\_{T_1+2}, \ldots, \hat{s}\_{T_1+T_2}]$. A predicted state is counted as valid only when its corresponding ground-truth was visited during the context period.
>
> For both one-step (Fig. 3A) and multi-step (Fig. 3B), all accuracies are computed as the fraction of correct predictions among these valid predictions.
>
>
> **10.  Unjustified Causal Claims from Correlational Data (Section 3.2, Figure 4)**
>
> We thank the reviewer for the careful review. We agree that our original claim is not rigor. We have revised it to the more cautious statement that:
>
> “these results suggest that the presence of grid-like cells is **highly correlated** with long-range prediction; one possible explanation is that stronger grid-like structure reflects better learning of spatial structures, which in turn supports improved prediction.”
>
> Finally, we appreciate the reviewer’s feedback, which helped us state our claims more precisely and rigidly.
>
> **11. Questionable Statistical Analysis in Figure 4A**
>
> We thank the reviewer for this helpful suggestion.
>
> We do not intend to argue that their relationship is "linear". To further clarify, in the revised manuscript, we added a Spearman correlation analysis **(Fig. A.8)**, which again shows a **strong monotonic relationship** between grid score and prediction accuracy ($ρ = 0.7833,\quad p = 10^{-6}$). To address concerns about uncertainty on the reviwer's request, we also include confidence interval(**Fig. A.8A**) and residual plots(**Fig. A.8B**).
>
> We have revised our claims in the main text to ensure they are stated more rigorously and conservatively, see reply in **Weakness 10**.
>
>
> **12. Undefined Error Metric and Inconsistent Terminology in 3D Experiments (Section 3.4, Table 1)**
>
> Thanks for pointing this out. We have refined our method to make the error metric and the 3D experiments clearer.
>
> **(1) Definition of the error metric and computation procedure.**
> Assume the input action–image observation sequence is
> $X_a = [(x_1, a_1), (x_2, a_2), \ldots, (x_T, a_T)]; x_t \in \mathbb{R}^{W \times H \times 3}$,
> and the predicted image sequence is
> $\hat{X} = [\hat{x}_2, \ldots, \hat{x}_T]; \hat{x}_t \in \mathbb{R}^{W \times H \times 3}$.
>
> The error metric is defined as the pixel-wise mean squared error (MSE), normalized by the total number of pixels:
>
> $$
> \text{error} = \frac{1}{3WH(T-1)} \sum_{t=2}^{T} |x_t - \hat{x}_t|_2^2.
> $$
>
> **(2) Typo correction**
>
> Thank you for pointing out the typo. We have corrected “accuracy” to “error” and added an explanation for the reported value of $std = 0.00$. This occurs because the actual standard deviation is extremely small and is rounded to zero in the table.
>
> **(3) Specification of the visible / non-visible criterion.**
>
> Our 3D environment has a size of $9\times 9$. For evaluation, we roughly discretize the environment into a $9 \times 9$ spatial grid. Since the network receives egocentric observations, we further discretize the orientation into 12 bins, resulting in $9 \times 9 \times 12 = 972$ distinct states. If an observation falls into a spatial grid and orientation that has been encountered previously, we classify it as **visible**; otherwise, it is classified as **non-visible**. Note that this discretization is used **only during evaluation**.
>
> (4) **Why is Hippoformer more stable?**
>
> Hippoformer achieves greater stability because the mm-TEM module organizes image representations with a structure-guided memory, where path integration provides a stable spatial scaffold combined with sensory features. This allows the model to encode observations coherently and capture spatial relationships between frames, enabling more consistent long-term predictions. In contrast, Transformer and Titan models lack such spatial priors, so their predictions deteriorate over long horizons due to a limited understanding of spatial structure.

---

> ### Author Response · Authors · 2025-11-22
> **Response to Reviewer 3ncY. Part 6**
>
> **13. Discussion Section Over claims Novelty and Omits Critical Limitations**
>
> We thank the reviewer for the insightful and responsible critique. We will address the issues regarding "overclaims" and the critical concerns point by point below.
>
> **(1) On “Overclaim of novelty on grid diversity”:**
>
> We have now cited and acknowledged the relevant prior work (Waniek, 2020; Stachenfeld et al., 2017; Dordek et al., 2016), clarified how our approach differs from these studies, and specified the new contributions introduced by mm-TEM. We also revised our claims to ensure they are precise and appropriately scoped. These updates are reflected in the revised manuscript (see **Weakness 8**).
>
> **(2) On “Foundational literature on HC–EC and factorized memory”:**
>
> We have revised the Introduction and Method to incorporate essential neuroscience and computational references (Eichenbaum et al., 2007; Hasselmo et al., 2002; Franzius et al., 2007; Bush et al., 2015; Stachenfeld et al., 2017). These revisions explicitly address the MEC–LEC functional dissociation and hippocampal binding roles (**Weakness 1**), acknowledge prior proposals of factorized sensory/structural systems, and situate our contributions relative to work linking prediction timescales with spatial scales. This provides a clearer placement of mm-TEM within the broader HC–EC framework.
>
> **(3) On “Methodological limitations”:**
>
> We thank the reviewer's attention to these issues and have clarified them as follows:
>
> - 2D generalization over novel transitions: We now clearly state that generalization concerns novel transitions (edges) between previously visited states, not unseen states (**Weakness 9**). Inferring these transitions remains non-trivial because the model must understand the underlying  spatial structure from only local sequential observations.
>
> - 64 step sequence coverage: 64 steps only cover a little fraction of transition edges to visited positions. As the context length or imagination length increases, the number of transition edges increases exponentially.
>
> - Factorization vs. direct associative memory: Ablations on Titans and positional embeddings have provided a direct associative memory baseline, and it shows that structural–sensory factorization combined with relational memory is required for strong performance (**reply in Weakness 6**).
>
> - Correlation vs. causation in grid–generalization analysis: All claims are now rephrased conservatively (“highly correlated”). We added Spearman correlation ($ρ = 0.7833, p = 10⁻⁶$) and residual plots to quantify uncertainty (**replies in Weakness 10 & 11**).
>
> - Clarification of 3D error metrics: Pixel-wise MSE is now explicitly defined, including the handling of visible and non-visible states (**reply in Weakness 12**).
>
> **(4) On “Efficiency and scalability claims”:**
>
> We refined our claims and updated analyses as follows:
>
> - Efficiency: We performed an additional experiment on TEM-t, a more efficient variant of TEM.  Wall-clock comparisons showing that mm-TEM achieves faster convergence and better long-range generalization than TEM-t. The complexity analysis is included in the revised manuscript (**reply in Weakness 6**).
>
> - Large-Scale Applicability: We have carefully refined our claims about large-scale environments, clarifying in the Discussion that testing performance in highly complex settings is reserved for future work.

---

> ### Author Response · Authors · 2025-11-22
> **Response to Reviewer 3ncY. Part 7**
>
> ### On Questions:
>
> 1. **Ablation of associative memory**
>
> We conducted new ablations comparing mm-TEM with a direct associative memory baseline (Titans) using the same meta-MLP architecture, without factorization, and clarified why warm-up and auxiliary losses are not applicable to such baselines (**Weakness 6**). Results confirm that factorized structural-sensory representations + relational memory are essential for multi-step prediction.
>
> 2. **Clarification of the 2D evaluation protocol**
>
> Accuracy is computed only for valid predictions, i.e., states visited during context (**Weakness 9**). Generalization is over unseen transitions (edges) rather than unseen states. Even with 64-step trajectories in 11×11 grids, many transitions remain unobserved, requiring inference of global spatial structure.
>
> 3. **mb parameter effect on grid scales**
>
> Ablations were performed separating training-time vs. test-time mb effects (**Weakness 7**). Results show that grid scale is determined by the path integration network itself; varying mb at test time without retraining does not change the emergent grid scale.
>
> 4. **Correlation analysis**
>
> We have added **Spearman correlation analysis (ρ = 0.7833, p = 10⁻⁶)**, residual plots, and discussion of monotonic relationships (**Weakness 11**). Also, we revised our claims carefully to reflect **correlation, not causation** (**Weakness 10**).
>
> 5. **Error metric in 3D space**
>
> Error metric: pixel-wise **mean squared error** normalized over all pixels (**Weakness 12**).
> Frames are classified as visible if the discretized position and orientation were previously encountered; otherwise not visible.
>
> 6. **Wall-clock time comparison**
>
> We have added wall-clock time comparisons with TEM-t, showing mm-TEM converges faster and achieves better generalization (**Weakness 6**, **Fig.A.6 in revised manuscript**). We temper scalability claims, noting that full large-scale performance requires further studies as illustrated in Limitations of the main text.
>
> 7. **Performance scalability to large environments**
>
> In 11x11 environment, 64-step context cover about 25% of states, details see **Weakness 9**.

---

> > ### Comment · Reviewer_3ncY · 2025-11-22
> >
> > I thank the authors for their thorough and thoughtful rebuttal. I was genuinely impressed by the care and completeness of it, not just with respect to my own review but also with respect to the other reviewers' comments. I appreciate the significant effort you invested in addressing each point. The clarifications to the methodology, the additional experiments and ablations, the improved empirical validation, and the strengthened positioning with respect to prior work have substantially improved the paper. Thank you for the considerable effort you put into these revisions. In light of these changes, I am happy to increase my score

---

> > > ### Author Response · Authors · 2025-11-22
> > >
> > > Thank you very much for your thoughtful and encouraging follow-up. We truly appreciate the time and care you devoted to reviewing our work and to providing detailed, constructive feedback. Your comments greatly helped us improve the clarity, rigor, and overall quality of the paper. We are grateful for your recognition of our revisions and for your updated assessment.

---

### Official Review · Reviewer_8yz6 · 2025-10-30

**Soundness:** 4
**Presentation:** 4
**Contribution:** 3
**Rating:** 8
**Confidence:** 4

**Summary:**

This work leverages recent formulations of memory from the Titan framework to improve models of the HPC-EC system. This in turn 1) improves the memory efficiency of the HPC-EC models over long contexts, 2) yields insights into grid scaling, 3) leads to an architecture that performs better than standard Titan and Transformer architectures. The new memory structure is an MLP which is trained in-context to map from keys to values, in contrast to older models based on Hebbian and more recently softmax attention.

**Strengths:**

This work is a significant and novel combination of recent ideas in neuroscience and machine learning. The improvement of tem-t using memory formulations from the recent Titans work is innovative and yields a fun result relating to grid scale. A hybrid architecture is proposed that blends the strengths of the titans-inspired model and transformer; however I would prefer it if the differential contributions of 1) the path-integration input (an old idea) and 2) the meta-MLP relational memory (new idea) were stated more clearly when comparing between models. The architecture has genuine promise to improve upon existing sequence models and I would be fascinated to see it deployed on language problems.

The figures are all clear and the text is very well written.

**Weaknesses:**

Barely a weakness & perhaps more to do with framing, but my understanding is that both Transformer and Titan control models you implemented do not have a recurrent path-integrator. Therefore the improvements demonstrated over each are primarily due to the path-integrator token input that the model receives. Tem-t also has this advantage and so should also perform similarly well, which should be mentioned. It would be nice to see more comparisons between tem-t and mm-tem. The use of the meta-MLP memory instead of softmax attention is elegant, I would be curious to see the gains explored slightly more. Perhaps MLP is more brittle than softmax when faced with a novel but semantically familiar input? Or perhaps MLP imbues memory retrieval with generalisation capacity since it is itself a NN? And does this impact the learnt g representations of the outer-loop meta network? Maybe the MLP allows more interesting operations to be done on memories, e.g. contextual splicing of memories - if memorised spatial envs A and B and then encounter env C in which half the stimuli are from A states and half from B

**Questions:**

Typos:
- 188, 194, 256, 735, 736

Do we have a curve for fig2B tem-t?

Can you clarify in main text that the transformer component of Hippoformer does not receive the path-integration input

Some questions about mb:
- What is the intuition for larger mb increasing the learnt grid size? A fun result that I don't fully understand!
- Does mb = 1 really perform that poorly - especially since there is no noise in the observations? If so then it would be nice to see this somewhere (since this is essentially the basis of the argument for including transformer in Hippoformer)
- Does smaller mb lead to better performance (assuming observations aren't noisy)? - If I understand correctly, a small mb ~ 1 is more similar to a transformer; would a network endowed with multiple memory MLPs each with different mb yield a similar performance to Hippoformer & also get grids at different scales?
- Linked to question above, could mb relate to oscillation frequency in HPC? I believe there is a dorsal-ventral gradient of oscillation frequencies in hippocampus (matches nicely with the gradient of grid scales). Also wonder if there are interesting findings relating to mb & the discretisation of grid scales that is observed.

Are there biological analogies to the hippoformer?

I've always been curious to see what the path-integrator module adds when applied to tasks that are less obviously cognitive map-like. Have you tried this model on text-based tasks?

Are there relations between the Titans meta-MLP and working memory (in contrast to transformer softmax which seems more episodic memory-like)? If so, maybe some kind of systems consolidation inspired ideas might apply here i.e. selectively exporting things out of hippocampal memory into neocortical memory

Line 453 - should transformers and mm-TEM be the other way round?

---

> ### Author Response · Authors · 2025-11-22
> **Response to Reviewer 8yz6. Part 1**
>
> We are deeply grateful for the reviewer’s generous evaluation and for the remarkably thoughtful and wide-ranging feedback. These questions not only clarified several subtle aspects of our design and analysis, but also opened genuinely exciting directions for future work. We appreciate the reviewer’s careful engagement with the manuscript and address each point in detail below.
>
> ### On weaknesses:
>
> **(1) Comparison between mm-TEM and TEM-t.**
>
> We conducted an additional experiment using TEM-t (**Appendix Fig. A.6**). As shown in **Fig. A.6C**, TEM-t converges substantially faster than the standard TEM—consistent with observations in [1,2]—making TEM-t a stronger baseline for evaluating mm-TEM.
>
> As our focus is on long-sequence generalization rather than performance within the training regime, we report multi-step imagination accuracy over gradient steps and wall-clock time (**Fig. A.6A–B**). Across both metrics, mm-TEM substantially outperforms the original TEM and TEM-t in training efficiency, demonstrating the advantage of incorporating a meta-MLP relational memory into the model.
>
> **Why TEM-t performs less well in long-sequence generalization.**
>
> This question requires our further investigation, but based on our current understanding, one factor may contribute: Non-parametric, attention-based memory management in TEM-t. TEM-t uses key-value cache as a non-parametric relational memory. And it relies on softmax attention for relational storage and retrieval. This memory management requires a hand-tuned similarity threshold to decide when to retrieve from memory. This fixed threshold may not adapt well across long sequences. In contrast, mm-TEM uses a **fully parametric meta-MLP relational memory**, where both storage and retrieval are data-dependent. This gives the system greater flexibility in how it allocates, retrieves, and composes relational information.
>
> **On whether an MLP is more brittle than softmax attention.**
>
>  In our current experiments, we did not observe such brittleness. Instead, as shown in **Fig. A.6A–B**, the **parametric** nature of the meta-MLP seems to enhance convergence on long-range relational inference.
>
> **On whether the MLP’s generalization affects the learned ( g_t ) representations.**
>
> Yes, our results in **Fig. A.6A–B** suggest a better generalization capacity in long-horizon prediction in mm-TEM, compared with TEM-t. In turn, it is able to shape the learning of the path-integration code $g_t$.
>
> **On the suggested experiment involving mixed environments A, B, and C.**
>
> We agree this is an interesting and valuable direction. Such mixed-environment task tests could probe how relational memory systems distinguish and recombine structural motifs. We plan to consider this in future work.
>
>
>
> ### On Questions:
>
> 1.  **Regarding “Typos”:** We thank the reviewer for pointing this out. We have corrected these typos in the revised manuscript.
>
> 2. **Regarding “TEM-t curve”:** Yes, we performed an additional control experiment of TEM-t. As shown in **Appendix Fig. A.6**, also see **our responses in Weakness** .
>
> 3. **Regarding “...clarify ... the transformer component of Hippoformer...”:** Thanks for pointing this out. In Hippoformer, Transformer do not recieve the input of path integration network, but the concatenation of action embeding and sensory embeding, the the concantenated embedding is added learnable positional embedding. We have clarified it in the revised manuscript.

---

> ### Author Response · Authors · 2025-11-22
> **Response to Reviewer 8yz6. Part 2**
>
> 4. **Regarding “What is the intuition for larger mb increasing the learnt grid size?”:**
>
> In mm-TEM, the meta-MLP memory is updated only once every mb time steps. Consider an input sequence  $[..., (g_t, s_t), (g_{t+1}, s_{t+1}), \dots, (g_{t+mb}, s_{t+mb}), ...]$. If the most recent memory update occurs at time $t$, then the next update will be at $t + mb$. The memory update at time $t + mb$ uses the batch  $[(g_{t+1}, s_{t+1}), ..., (g_{t+mb}, s_{t+mb})]$. This means that during the interval $t \rightarrow t+mb$, the network does not yet have access to these future $(g,s)$ pairs. All predictions made in this interval must rely solely on information available at or before time $t$. As a result, the network must effectively predict 1 to $mb$ steps in the future. A larger value of $mb$ therefore imposes a longer prediction horizon.
>
> Why a Large Prediction Horizon Encourages Large Grid Scales:  An possible intuitive explanation is as follows. In mm-TEM, the relational memory stores historical $(g, s)$ pairs. Predictions are produced by using the structural code $g$ to query the relational memory and retrieve the associated sensory code $s$.
>
> For **small grid scales**, the structural code is highly precise, but grid codes of neighboring locations are **weakly correlated**. This makes the representation more sensitive to accumulated errors from path integration.
>
> For **large grid scales**, the grid codes are less spatially precise, yet **more strongly correlated** across nearby positions. Each code carries richer contextual information about its surroundings and is more robust to accumulated integration error.
>
> Consequently, when the prediction horizon is short (e.g., $mb = 1$), accumulated path integration error remains small, so a highly precise spatial code is advantageous. However, when the prediction horizon is long (e.g., $mb = 8$), accumulated error grows because sensory feedback for error correction is sparse (only every $mb$ steps). Under these conditions, a more correlated and robust representation becomes beneficial, making **larger-scale grid codes** preferable during optimization.
>
> 5. **Regarding “Does mb = 1 really perform that poorly ... ”:**
>
> In the 2D grid task, following the setup of James C. R. Whittington et al. [1], there is no observation noise. As shown in **Appendix Fig. A.2**, mm-TEM exhibits a trade-off as $mb$ varies. With a small $mb$ (e.g., $mb=1$), the model generalizes well for short-range prediction (**Fig. A.2A**) but performs worse on long-range prediction compared to the $mb = 4, 8$ settings (**Fig. A.2B**) . Conversely, a large $mb$ (e.g., $mb=8$) improves long-range prediction while reducing short-range accuracy.
>
> Why does mm-TEM with (mb = 1) perform worse than (mb = 8) in long-range prediction? An intuitive explanation is as follows. Although the 2D grid task contains no observation noise, the transformation from action input to structural code in the path-integration network is not perfect. Small errors or distortions accumulate during multi-step imagination when sensory error correction is absent. With $m = 1$, the grid scale is small, providing high spatial resolution but weak correlations across nearby locations, which makes the structural code highly sensitive to accumulated error - resulting in strong short-range but poor long-range prediction. In contrast, with $m = 8$, the grid scale is larger, yielding lower spatial resolution but stronger location correlations and greater robustness, leading to weaker short-range but better long-range performance.
>
> 6. **Regarding “does smaller mb lead to better performance ... ”**
>
> As noted above, the 2D grid task contains no observation noise. A small $mb$ is good at short-range prediction compared to large $mb$, much like the behavior of a transformer with a limited context window.
>
> Extending mm-TEM to incorporate multiple memory MLPs with different $mb$ values is indeed an interesting idea. In principle, equipping the network with several memory MLPs - each operating at a distinct $mb$ - could enable it to match Hippoformer’s performance and generate grid representations at multiple spatial scales.
>
> In my view, the key difference between using multiple memory MLPs and the Hippoformer architecture lies in **training efficiency**. Hippoformer combines a Transformer with a limited context window and an mm-TEM module with a large $mb=8$. A large $mb$ enables the memory MLP to support fast, highly parallelizable training [3], which can improve Hippoformer’s scalability compared to a design that relies on multiple memory MLPs. This improved scalability makes Hippoformer a more promising architecture for extending hippocampus-inspired models to more complex real-world settings, such as video or text domains.

---

> ### Author Response · Authors · 2025-11-22
> **Response to Reviewer 8yz6. Part 3**
>
> 7. **"Regarding “ ... Could mb relate to oscillation frequency in HPC ? .... relating to mb ... grid scales ..."**
>
> The idea is highly intriguing because oscillation frequency is suggested to modulate Hebbian learning[4, 5], providing a natural mechanism to influence memory update and learning in the biological brain. Additionally, hippocampal oscillations at multiple frequencies have been observed to exist along the AP axis and function distinctly in spatial navigation[11]. Such gradient is also found in molecular level to informs behavioral systems[12].
>
> How discretisation of grid scales emerges is an interesting open question[6]. At present, our model does not exhibit clear evidence that relates $mb$ to the discretization of grid scales. One possible way to investigate this is to extend mm-TEM with multiple memory MLPs as mentioned in the reviewer’s suggestion, each equipped with an adaptive, learnable $mb$. This design may allow the model to autonomously learn different **$mb$** values, enabling us to investigate the emergence of grid scale discretization. We leave this direction for future work.
>
> 8. **Regarding “are there biological analogies to the hippoformer ? ”**
>
> Yes, one possible biological analogy to Hippoformer is the interacting PFC–HPC system. Experimental evidence suggests that neurons in the prefrontal cortex and hippocampus cooperate to maintain and manipulate information in memory. The hippocampus is primarily responsible for storing and organizing memories, while the prefrontal cortex contributes to attention and control the hippocampal memory [7, 8]. In Hippoformer, the Transformer component may serve as a working-memory–like system for control and attention, whereas the mm-TEM module plays a hippocampus-like role by storing and maintaining memory representations.
>
> 9. **Regarding"... what the path-integrator module adds ..., when ... less obviously cognitive map-like ..."**
>
> We have not yet tested our model on text-based tasks, this is an exciting direction for future work. Notably, the idea of linking hippocampal cognitive map theory to language or text has a long intellectual history. In _The Hippocampus as a Cognitive Map_, O’Keefe proposed that **“the cognitive-mapping system could function as a deep structure for language,”** where “deep structure” refers to the syntactic structure of language.
>
> Recent experimental and theoretical developments have brought renewed attention to this question[9,10]. For example, Whittington et al. suggested that a velocity-driven grid system could represent syntactic structure via a form of path integration[9] - analogous to spatial navigation - and recent computational work has begun to support this hypothesis[10]. Exploring mm-TEM in language domains may therefore provide a promising avenue to further connect hippocampal-inspired structure learning with linguistic abstraction.
>
> 10. **Regarding "... relations between the Titans meta-MLP and working memory ..."**
>
> Thank you for the insightful suggestions. In the current Hippoformer, the meta-MLP memory operates in parallel with the Transformer-like working memory. However, in biological systems, the interaction between prefrontal working memory and hippocampal episodic memory is far richer[7,8] and plays a central role in consolidation[13] and schema formation[14]. The reviewer’s suggestion - such as selectively exporting information from hippocampal memory into neocortical memory - highlights exactly the kind of bidirectional coordination that our current architecture does not yet model. These biological interactions provide valuable inspiration for designing more sophisticated mechanisms of communication between the working-memory module and the structural-memory module, and we consider this an exciting direction for future development.
>
>
> 11. **Regarding "Line 453 - should transformers and mm-TEM be the other way around."**
>
> Line 453 — _“transformers contribute to abstraction, whereas mm-TEM focuses on memorization”_ is correct. In our formulation, mm-TEM functions essentially as a grid-structured, scaffolded memory system.
>
>
> ### References
>
> [1] James C.R. Whittington et al., Cell, 2020
>
> [2] James C.R. Whittington et al., ICLR, 2021
>
> [3] Ali Behrouz et al., NeurIPS, 2025
>
> [4] TM George et al., eLife, 2023
>
> [5] Xiaojing wang et al., Physiological reviews, 2010
>
> [6] Mikail Khona et al., Nature, 2025
>
> [7] Timothy Spellman et al., Nature, 2015
>
> [8] Jonathan Daume et al., Nature, 2024
>
> [9] JCR Whittington et al., Nature Neuroscience, 2022
>
> [10] XL Zou et al., ICML, 2024
>
> [11] Abhinav Goyal, et al, Nat Comm, 2020
>
> [12] Vogel, J.W., et al, Nat Comm, 2020
>
> [13] Jens G. Klinzing et al., Nature Neuroscience, 2019
>
> [14] Asaf Gilboa et al., Trends in Cognitive Sciences, 2017

---

### Official Review · Reviewer_Ny5p · 2025-11-01

**Soundness:** 3
**Presentation:** 2
**Contribution:** 2
**Rating:** 4
**Confidence:** 5

**Summary:**

This paper introduces Hippoformer, a hybrid architecture combining a novel hippocampus-inspired memory module, mm-TEM, with a Transformer. The work aims to address the lack of inherent spatial priors in standard Transformers. The central contribution lies in mm-TEM, a scalable variant of the Tolman–Eichenbaum Machine that employs a meta-MLP for relational memory, closely resembling the long-term memory module used in Titan. The authors conduct experiments in 2D and 3D environments, reporting that Hippoformer achieves superior generalization on long-horizon spatial prediction tasks compared to Transformer and Titan baselines.

While the paper is well-written and tackles an important problem, I have major concerns regarding the experimental methodology and the clarity of the architectural contributions. The central claims about the superiority of the proposed relational memory are not sufficiently supported because the baseline comparisons appear to be confounded by critical differences in how positional information is handled.

**Strengths:**

* **Strong Motivation**: The paper is motivated by a clear and significant limitation of current generative models—their lack of structured spatial memory. The inspiration drawn from the hippocampal-entorhinal system provides a principled foundation for the architectural design.
* **Interesting Component Design**: The mm-TEM module, with its meta-MLP and auxiliary relational losses, is an interesting and efficient take on prior hippocampus models. The analysis showing the emergence of grid-like representations is a valuable piece of evidence supporting the design.

**Weaknesses:**

The paper's central claims hinge on the superior performance of mm-TEM and Hippoformer over strong baselines, particularly in length generalization. However, the experimental setup is insufficiently described and may contain confounding variables that invalidate these conclusions.

1.  **Lack of Clarity on Positional Encoding in Baselines:** The most critical issue is the ambiguity surrounding the positional encoding (PE) used for the Transformer and Titan baselines. The mm-TEM model relies on a recurrently updated structural code, `g_t`, from a Path Integration Network, which effectively serves as a powerful, dynamic, and task-specific form of positional encoding. The paper does not specify whether the baselines have access to this same structural code.
    *   If the baselines use standard, fixed positional encodings (e.g., sinusoidal) or no PE at all, the comparison is fundamentally flawed. The performance gains of Hippoformer could stem entirely from its superior positional information, not its relational memory. As shown by Kazemnejad et al. (NeurIPS 2023, "The Impact of Positional Encoding on Length Generalization in Transformers"), the choice of PE is a dominant factor in a Transformer's ability to generalize to longer sequences. To isolate the contribution of the relational memory, the baselines must be equipped with a similarly powerful and dynamic PE.
    *   This lack of clarity makes it impossible to attribute the performance gains to the claimed source (the relational memory) versus a known, powerful factor (the positional encoding scheme).

2.  **Unclear Architectural Novelty Compared to Titans:** The paper presents Hippoformer as a combination of a Transformer and the mm-TEM module (which is centered around a meta-MLP memory). The Titan architecture is also described as a model leveraging fast MLP weights. From the descriptions provided, the high-level architectural blueprint of Hippoformer appears very similar to that of Titans. The paper needs to explicitly detail the architectural and mechanistic differences. Is the primary novelty of Hippoformer simply the introduction of the auxiliary relational loss as a form of inductive bias for the meta-MLP? If so, the contribution should be framed more narrowly as a novel training objective for existing hybrid architectures on spatial tasks, rather than a fundamentally new architecture.

3.  **Ambiguity in the Training of the Recurrent Module:** The mm-TEM module contains a recurrent update for the structural code `g_t`. This introduces dependencies across the entire sequence. The paper lacks crucial details about how this recurrence is handled during training. Is backpropagation through time (BPTT) performed over the full sequence length? Or is it truncated? This detail has significant implications for the model's computational cost, memory requirements, and its practical ability to capture the long-range dependencies it is being credited for.

**Questions:**

**1. Clarifications on Baseline Models (Crucial for Rebuttal):**
This is the most important area. A clear response here could significantly change my assessment.

*   **Question 1a (Positional Encoding):** How was positional/structural information provided to the Transformer and Titan baseline models? Specifically, did they take as input only `[s_t, a_t]`, or did they also receive the structural code `g_t` from the Path Integration Network, similar to mm-TEM?
*   **Question 1b (Type of PE):** If the baselines did *not* use the Path Integration Network, what form of positional encoding was used (e.g., sinusoidal, learned, rotary, or none)?

**2. Architectural and Contribution Framing:**

*   **Question 2 (Hippoformer vs. Titans):** Could you please provide a more detailed, side-by-side comparison of the Hippoformer and Titan architectures? What are the key differences in their memory update rules, the interaction between the MLP-based memory and the self-attention component, and the flow of information? A diagram or table would be very helpful.

**3. Implementation Details for Reproducibility and Analysis:**

*   **Question 3 (Training Recurrent Module):** How is the gradient calculated for the recurrent mm-TEM module? Is backpropagation through time (BPTT) applied over the full sequence length (e.g., 256 steps in Fig. 2), or is it truncated to a smaller window? What are the implications of this choice for computational complexity and memory usage during training?


**Suggestions**
* If the Fig. 3A experiment corresponds to the length generalization task and does not use the Path Integrator Network for g_t or employ g_t as a positional encoding, the authors should evaluate multiple positional encoding methods and compare their results to those of mm-TEM.

---

> ### Author Response · Authors · 2025-11-22
> **Response to Reviewer Ny5p. Part 1**
>
> We sincerely thank the reviewer for the time and effort devoted to evaluating our submission. The comments raised are constructive and highly valuable for improving the clarity and rigor of the work. We have carefully addressed each point with additional clarifications, ablations, comparisons and revisions to the manuscript. We believe these have substantially strengthened the work.
>
> ### **On Weaknesses:**
>
> ###  **Weakness 1: Lack of Clarity on Positional Encoding in Baselines**
>
> We thank the reviewer for raising this important point about positional encoding (PE). We agree that PE is a critical factor in Transformer generalization and have therefore clarified our methods and added new ablations to strength our claims.
>
> **(1) Clarification of PE in baselines (now stated clearly in Method).** In all original experiments (2D and 3D), Transformer and Titan baselines used learned axial positional embeddings(Kitaev et al., 2020), a common choice in long-sequence Transformers, which we refer to as "learned" throughout. This is now explicitly stated in the revised manuscript.
>
> **(2) New PE ablation/control experiments (added to Appendix A.7).** To determine whether mm-TEM/Hippoformer’s advantage stems from hippocampus-like computational mechanism, we performed additional experiments and trained Transformer/Titan baselines with three PE variants:
> - **Sinusoidal PE** (static, absolute)(Vaswani et al., 2017)
> - **Rotary PE** (relative) (Su et al., 2024)
> - **Dynamic PE**, implemented by a path integation network (PIN) identical to mm-TEM’s PI module, serving as a powerful recurrent positional embedding.
>
> We evaluated both one-step and multi-step prediction and also tested transfer in the circular-grid environment (see Fig. A.4).
>
> **(3) Key results (see Appendix Fig. A.3 and Fig. A.4 in the revised manuscript).**
>
> - Rotary PE and dynamic PE substantially improve one-step prediction, in some cases reaching performance comparable to mm-TEM with ($mb=1$), see **Appendix Fig. A.3A**.
>
> - However, none of these PE variants allow Transformers or Titans (Behrouz et al., 2024) to approach mm-TEM’s performance on multi-step prediction (see **Appendix Fig. A.3B**.) or circular-grid transfer, both of which require robust long-range relational inference (see **Appendix Fig. A.4AB**).
>
> - Hippoformer (Transformer + mm-TEM with $mb=8$) consistently outperforms all PE baselines by a wide margin, demonstrating the complementary strengths of short-range memory (Transformer) and long-range, structure-aware memory (mm-TEM).
>
> - Importantly, mm-TEM and Hippoformer achieve these results with fewer parameters than the Transformer and Titan baselines (**see Appendix Tab. A.2**), indicating that the gains are not due to model size, but architecture.
>
> **(4) Why PE alone cannot explain the gains.** Strong positional encodings (rotary or dynamic PI-based) improve short-range prediction, but they do not endow baseline Transformers/Titans with hippocampus-like computational mechanisms - such as feedback-based error correction or relational memory. Simply inserting a recurrent PI module as a positional embedding does not guarantee hippocampus-like abstract structure learning within the PI module, which are essential for multi-step relational reasoning and long-range prediction.
>
> Overall, the additional ablations confirm the clear performance advantages of mm-TEM and Hippoformer over strong PE baselines.

---

> ### Author Response · Authors · 2025-11-22
> **Response to Reviewer Ny5p. Part 2**
>
> #### **Weakness 2: Unclear Architectural Novelty Compared to Titans**
>
> Thank you for raising this important point. We would like to state that Hippoformer is _not_ a variant of Titans, nor is its novelty limited to the auxiliary relational loss. In contrast, they differ substantially in the memory systems, integration strategies and computational mechanisms, as detailed below.
>
> **(1) Memory system: MEC–HC–inspired structured spatial memory in mm-TEM vs. flat fast-weight associative memory in Titans**
>
> - In mm-TEM, the meta-MLP memory trained with an auxiliary relational loss functions as a relational memory. Together with the feedback-based error-correction from memory to the path-integration network, mm-TEM implements a minimal MEC–HC–like system. This architecture benefits from an explicit structure–content factorization, enabling the model to extract abstract spatial structure and transition rules (as reflected in the grid patterns in Fig. 2 and Fig. 4). Within Hippoformer, mm-TEM therefore supports structure-scaffolded memory organization.
>
> - In contrast, the meta-MLP memory in Titans is deeply entangled with the Transformer backbone and lacks any structure–content separation. It functions essentially as a flat fast-weight associative memory, without the inductive bias or mechanism needed for learning abstract spatial structure. Consequently, it performs context-dependent memorization, contrasting with mm-TEM's approach of memory organization guided by spatial structure.
>
>  **(2) Integration of long-term and short-term memory**
>
> While both Hippoformer and Titans share the conceptually similar idea of leveraging the complementary strengths of short-term (Transformer) and long-term (meta-MLP) memory, Hippoformer employs a distinct integration architecture that more clearly demonstrates these complementary strengths and their resulting benefits.
>
> - In architecture, Hippoformer places the structured memory and transfomer pathway _in parallel_, each contributing complementary predictions: the Transformer supports short-horizon prediction, while mm-TEM acts as a structural spatial memory enables long-horizon and structural generalization. In Titans, long-term memory (meta-MLP) interacts with transformer in a sequential and entangled manner.
>
> - Hippoformer demonstrates clear complementary behavior (**Fig. 5, Appendix Fig. A.2**), which is absent in Titans. We established in mm-TEM that the memory update frequency, governed by $mb$, produces a trade-off in performance(see **Appendix Fig. A.2**): a larger $mb$ enhances long-horizon prediction and computational efficiency but degrades short-horizon prediction. A smaller $mb$ yields the opposite outcome.
>
> Hippoformer successfully integrates these findings by pairing a context-limited Transformer for short-term prediction with an mm-TEM module (set with a larger $mb$) for learning long-term predictive structure. This combined approach achieves good prediction accuracy at both horizons while simultaneously improving training efficiency.
>
>
> #### **Weakness 3: Ambiguity in the Training of the Recurrent Module**
>
> Thank you for highlighting this important point. We have now clarified the details in the manuscript and added new control experiments.
>
> (1) For all primary results, models are trained on sequence lengths of 128 or 256 using full BPTT without truncation.
>
> (2) To examine whether full BPTT is necessary, we conducted a new experiment with 512-length sequences (see **Appendix Fig. A.5**).
>
> Hippoformer: BPTT truncated at 256 steps.
>
> Transformer baseline: full BPTT over all 512 steps.
>
> Despite the truncated training, Hippoformer still outperforms the Transformer, demonstrating that full BPTT is not required for Hippoformer to capture long-range dependencies.
>
> (3) Long-range dependency does not rely solely on long unrolled recurrence. It shows that learning long-range dependency also relies on the architectural prior to learn abstract structural codes, rather than solely depending on brute-force recurrence over hundreds of steps.
>
> (4) Efficient recurrence in the PI module. The path-integration network is small, and in the future we plan to explore parallelizable linear RNN architectures (e.g., Mamba-type models). In the current design, feedback from the relational memory to the PI network occurs every $mb$ steps, providing effective error correction while greatly reducing training cost.
>
> We have added the truncated-BPTT ablation in the revised manuscript (**Appendix Fig. A.5**) and clarified the recurrence handling in the Method.

---

> ### Author Response · Authors · 2025-11-22
> **Response to Reviewer Ny5p. Part 3**
>
> **On Questions**:
>
> #### **Question 1: Clarifications on Baseline Models**
>
> a) For all Transformers and Titans baselines, the input consists only of $[s_t, a_t]$. They do not receive the structural code $g_t$ from the PI Network. The use of $g_t$ is unique to mm-TEM and Hippoformer, and is essential to their structured memory mechanism.
>
> b) All baselines used learned positional embeddings: (1) $a_t$ is linearly embedded and concatenated with $s_t$; (2) A learned positional embedding is then _added_ to the resulting vector before entering the Transformer or Titan layers.
>
> To ensure fairness, we conducted new experiments (2D grid) where baselines were equipped with alternative positional encodings, including _sinusoidal PE, rotary PE, and dynamic PE_ analogous to mm-TEM  (details in **“Weakness 1”** and **Appendix Fig. A.3，Fig.A.4**).
>
> Across all variants, mm-TEM and Hippoformer consistently outperform the baselines, particularly in multi-step prediction tasks that require understanding the underlying spatial structure and relational knowledge.
>
> We have now clearly documented the exact positional encoding used for each baseline and the results of the new control studies in the revised Method and Appendix.
>
> #### **Question 2: Architectural and Contribution Framing**
>
> We appreciate the suggestion. A direct comparison is important, and we provide a side-by-side table to address this as shown below.
> Hippoformer employs **the same novelty-driven meta-MLP update rule** as Titans, as this approach is consistent with hippocampal memory update characteristics.
> And we summarize the differences as below.
>
> |                          **Aspect**                           | **Hippoformer**                                                                                                                                                                           | **Titans**                                                                                                                                                                     |
> | :-----------------------------------------------------------: | :---------------------------------------------------------------------------------------------------------------------------------------------------------------------------------------- | :----------------------------------------------------------------------------------------------------------------------------------------------------------------------------- |
> |                   **Long-term Memory Type**                   | 1. Structured spatial memory (mm-TEM) inspired by MEC–HC architecture. 2. meta-MLP functions as a **relational memory**. 3. Explicit **factorized structure–content** code. | 1. No structural prior. 2. meta-MLP behaves as a **context-dependent associative memory**.  3. Memory is **entangled** and lacks structure–content factorization. |
> |                     **Positional System**                     | 1. **Dynamic**, action-dependent positional system with learned path integration. 2. be able to encode spatial rules (e.g., "$south + north + west + east = \mathbf{0}$").         | 1. Static positional encodings (learned or rotary), not action-dependent 2.  cannot encode spatial update rules.                                                        |
> | **Interaction Between Positional System and meta-MLP Memory** | Explicit **feedback / error-correction pathway** from relational memory to the positional system.                                                                                         | No interaction between positional encodings and the meta-MLP memory.                                                                                                           |
> |                    **Macro Architecture**                     | mm-TEM and Transformer operate **in parallel**, combining predictions at the output; effectively leverages complementary short-/long-range memory.                                        | Memory is **interleaved** with Transformer layers; no clear evidence of complementary short-/long-range memory specialization.                                                 |
> |                  **Generalization Ability**                   | Learns **abstract spatial structure** and supports robust **long-range prediction**.                                                                                                      | Fails to acquire abstract spatial structure; poor long-range prediction.                                                                                                       |
>
>
> ####  **Question 3: Implementation Details for Reproducibility and Analysis"**
>
> Thank you for pointing this out. We have updated the training details in the revised manuscript. For all main experiments, BPTT is performed over the entire sequence (no truncation). For longer sequences, BPTT can be truncated to reduce computational and memory cost. See **Weakness 3** for details.

---

> > ### Author Response · Authors · 2025-11-22
> > **Response to Reviewer Ny5p. Part 4**
> >
> > On Suggestions:
> > 1. Thank you for the helpful suggestion. We have incorporated the recommended experiments in the revised manuscript; please see **Weakness 1** and the response to **Question 1** for details.
> >
> > References:
> >
> > [1] Kitaev et al., arXiv, 2020
> >
> > [2] Vaswani et al., NeurIPS, 2017
> >
> > [3] Su et al., Neurocomputing, 2024
> >
> > [4] Behrouz et al., arXiv, 2024

---

### Official Review · Reviewer_Be6v · 2025-11-01

**Soundness:** 3
**Presentation:** 3
**Contribution:** 3
**Rating:** 6
**Confidence:** 3

**Summary:**

This paper addresses the limitations of prior hippocampal-inspired models (TEM's computational inefficiency) and modern architectures (Titans' lack of inherent spatial memory). By synthesizing these approaches, the authors propose **mm-TEM** (meta-MLP TEM), demonstrating better training efficiency and revealing a novel link between the memory update frequency and the emergence of biologically meaningful grid-like representations. Furthermore, they introduce **Hippoformer** (mm-TEM + Transformer), a hybrid architecture that effectively integrates structured long-term spatial memory with precise short-term working memory, achieving robust generalization across demanding 2D and 3D prediction tasks.

**Strengths:**

*   **Clarity and Organization:** The paper is well-structured with clear conceptual figures (e.g., Figure 1), making the overall architecture and training rationale highly intuitive.
*   **Compelling Rationale:** The core rationale for the proposed method—integrating the computational efficiency of the meta-MLP memory (inspired by Titans) into the theoretically grounded TEM framework—is valid for overcoming scalability issues while retaining biological plausibility.
*   **Strong Experimental Validation:** The systematic evaluation, including ablations and generalization tests across long context, multi-step imagination, and distribution shifts (circular-grid), robustly supports the claim that mm-TEM captures underlying spatial structure more faithfully than baseline models.

**Weaknesses:**

**Literature Review Suggestion (Line 40):** The Introduction's discussion of the Transformer's associative memory perspective would be strengthened by citing recent, relevant works that formalize the Transformer's components (e.g., FFNs) as explicit memory systems, such as:
- Geva, Mor, et al. (2020) on FFNs as key-value memories.
- Ramsauer, Hubert, et al. (2020) on Hopfield networks' relation to attention.

**Relational Loss Notation and Rationale:**
- **Notation Clarity:** The auxiliary relational losses (Lines 184-190) require precise clarification. Please map the main text's definitions\
$L_1$ and $L_2$ to the figure notations of $L_{x2g}$, $L_{g2x}$, $L_{g2g}$ and clarify the missing one.

- **Missing Term Rationale:** The authors can define four potential combinations ($g \to x$, $x \to g$, $g \to g$, $x \to x$). But figure/main text uses 2/3 losses. What is the underlying reason for excluding the other possible terms from the relational losses?

2.  **Path Integration Network Ablation and Role:**
    *   The core TEM theory mandates a Path Integration (PI) network for structural code generation. Since the authors emphasize the **novel meta-MLP relational memory** as the main source of generalization, could the authors include an **ablation study on the PI network component itself** (e.g., replacing it with a simpler, non-integrated recurrent mechanism or removing the error correction loop)? This would help quantitatively disentangle the contribution of the *novel memory* from the *PI component*.
    *   Given the PI network's role in generating the structural code ($g_t$) from actions, could this component be viewed as an advanced form of **learned positional encoding**?

3.  **Grid Pattern Neuron Selection:**
    *   For the analysis of emergent grid-like representations (Figure 2C, Figure 4B), the authors show visualizations of the high-gridness neurons. Could the authors explicitly state **from which module** (the Path Integration Network or the Relational Memory Network) the "Top-5/Top-3" high-gridness neurons were selected? Clarification on the specific origin of these analyzed neurons is crucial for interpreting the results.

4.  **Table 1 Completeness (3D Task):**
    *   Table 1 presents results for Transformer, Titans, and Hippoformer in the 3D environment. Given that **mm-TEM** is the core component providing the long-term generalization in Hippoformer, why are the performance results for the **mm-TEM** model alone **omitted**? (Figure 5 shows on par imagination capability between mm-TEM and hippoformer) Including mm-TEM's 3D performance would provide a necessary direct measure of the Transformer's specific contribution (abstraction) to the final Hippoformer architecture in this complex domain, thereby strengthening the claim of synergy.

**Questions:**

See weakness

---

> ### Author Response · Authors · 2025-11-22
> **Response to Reviewer Be6v. Part 1**
>
> We thank the reviewer for the insightful and focused feedback. The issues highlighted have led us to improve both the presentation and the empirical support of the work, including added clarifications and targeted ablation studies.
> We address each comment in detail below and have updated the manuscript to reflect these improvements.
>
> #### **On Weaknesses**:
>
> **Weakness 1. Regarding "Literature Review Suggestion":** Thank you for the suggestion. We have added the recommended citations ([1], [2]) to strengthen the discussion of the transformer’s associative-memory perspective in the introduction.
>
> **Weakness 2. Regarding "Relational Loss Notation and Rationale":**
>
> 1. Thanks for pointing this out. $L_1$ refers to $L_{x2g}$ , $L_2$ refers to $L_{g2g}$ . We have revised these terms **in Lines 192-201** of the revised manuscript as follows:
>
> To explicitly enforce relational binding, we introduce three auxiliary  relational losses (Fig.1D):
>
> (1) Structure from content: retrieve  $\hat{g}\_{t}$ given only  $x_t$ ($m_t = [\mathbf{0}, x_t]$), minimized by $\mathcal{L}_{x2g} = \| \hat{g}_t - g_t \|_2^2$.
>
> (2) Structure from structure : retrieve  $\bar{g}\_t$ given only $g_t$  ($m_t = [g_t, \mathbf{0}]$), minimized by $\mathcal{L}_{g2g} = \| \bar{g}_t - g_t \|_2^2$.
>
> (3) Content from the structure: retrieve $\hat{x}\_t$ given only $g_t$ ($m_t = [g_t, \mathbf{0}]$), minimized by $L_{g2x}=\|\hat{x}_t-x_t|^2_2$.
>
> The total relational loss is $L\_{\text{rel}} = L_{x2g} + L_{g2g} + L_{g2x}. $ Since $L_{g2x}$ is accounted for in the main predictive-learning objective, it can be absorbed into that term. Thus, the relational loss can be simplified to $L_{\text{rel}} = L_{x2g} + L_{g2g}.$
>
> 2. We conducted an additional ablation study, **reported in Fig. A.9 of the revised manuscript**. The results show that adding $L_{x2x}$ to the auxiliary relational loss slightly reduces performance on both one-step and multi-step predictions.
>
> In the original formulation $L_{\text{rel}} = L_{x2g} + L_{g2g}+L_{g2x}$ ,  both components play essential roles in mm-TEM. $L_{x2g}$ and $L_{g2g}$ are crucial for the error-correction process: the model must use both sensory and structural codes to extract the correct structural code into relational memory, which is then passed to the path integration network to recalibrate accumulated errors.
>
> $L_{g2x}$ is indispensable for prediction, the model uses the structural code $g$ from the path integration network to query relational memory. In contrast, $L_{x2x}$ does not contribute to either the error-correction process or the prediction process, which explains why its inclusion does not improve - and in fact slightly degrades - overall performance.
>
> **Weakness 3. Path Integration Network Ablation and Role**
>
> 1. Thank you for the constructive comments. We conducted an additional ablation study in which we removed the error-correction loop, making the path-integration network non-integrated. The results **(Fig. A.10)** show that removing the error-correction path leads to a dramatic drop in performance.
>
> In mm-TEM, both the path-integration network and the relational memory are essential. A well-formed relational memory enables accurate relational retrieval, which in turn supports structure learning in the path-integration network. Conversely, a well-organized path-integration code provides reliable structural/positional representations that facilitate memory organization in mm-TEM.
>
> In this ablation, the error-correction pathway mediates the interaction between relational memory and the path-integration network during both training and testing. When this pathway is removed, the model struggles to form meaningful structural codes, leading to degraded overall performance.
>
> 2. Yes, the PI network can be viewed as an action-dependent learned positional encoding, as having been proposed in TEM-t [3,4].
>
> Unlike static encodings such as sinusoidal or rotary positional encodings, a PI-like positional encoding can capture spatial structure and transition rules - for example, relations such as _north + west + south + east = 0_ - which supports structure-based inference. However, this PI-like encoding cannot arise in isolation. The structural regularities encoded by the PI network are learned through its interaction with the relational memory, rather than being hard-coded.
>
> Together, the PI-based positional encoding and the relational memory system enable the model to compress long spatial sequences in an ordered and structurally meaningful way.
>
> In this work, our goal is to incorporate these hippocampal-inspired mechanisms into deep learning architectures to improve their ability to handle long sequences.

---

> ### Author Response · Authors · 2025-11-22
> **Response to Reviewer Be6v. Part 2**
>
> **Weakness 4. Grid Pattern Neuron Selection**
>
> The "Top-5/Top-3" high-gridness neurons were selected exclusively from the PI network, which models the biological medial entorhinal cortex (MEC), where periodic spatial representations (grid-like patterns)—are canonically found. The meta-MLP memory, which models the hippocampal system, then uses these structural codes for relational binding sensory codes.
>
> We have clarified the origin of these neurons in the revised manuscript (e.g., in the captions for Fig. 2C and Fig. 4B, and the relevant Methods section) to ensure precise interpretation of the results.
>
> **Weakness 5. Table 1 Completeness (3D Task)**
>
> 1. We thank the reviewer for the suggestion. We have added the mm-TEM 3D performance, trained under the same protocol as Hippoformer, **see Tab.1 in the revised manuscript.**
>
> Note that the transformer in Hippoformer uses only a single layer and a limited context window of 32, and by itself cannot perform accurate prediction in the 3D task. The results confirm that Hippoformer benefits from the **synergistic interaction** between the transformer and mm-TEM.
>
>
> **References:**
>
> [1] Geva, Mor, et al. _EMNLP_. 2021.
>
> [2] Ramsauer, Hubert, et al. _arXiv_ 2020.
>
> [3] James C.R. Whittington et al., ICLR, 2021
>
> [4] James C.R. Whittington et al., Nature Neuroscience, 2022

---

> ### Comment · Reviewer_Be6v · 2025-11-25
>
> Thank you for the response.
>
> One more question: Regarding the updated table 3D task for mm-TEM, I find mm-TEM only is good at Imagination but not for single step prediction.
> Can you explain/discuss about potential reason for this?
>
> (Minor: Reference [3] is James C.R. Whittington et al., ICLR 2022, not 2021 I guess...?)

---

> > ### Author Response · Authors · 2025-11-26
> >
> > 1. Thank you for the comment. This is indeed an important question.  **Short answer:** both mm-TEM alone and mm-TEM inside Hippoformer were trained with **$mb = 8$** on the 3D task. **mm-TEM trained with a large mb (e.g., $mb=8$) is inherently biased toward long-horizon prediction, which improves multi-step imagination but weakens single-step accuracy.**
> >
> >
> > Below we provide a more detailed explanation.
> >
> >  (1) **Why this bias exists**. The key reason is that the **mm-MLP memory is updated only once every $mb$ steps**. Consequently, during training the model implicitly learns to predict over a horizon of $mb$ future steps.
> > - Large $mb$ (e.g., $mb=8$): The model must use information from the last memory update to make predictions up to $mb$ steps ahead. This encourages learning representations and computations that are robust to multi-step error accumulation - i.e., better long-range prediction.
> > - Small $mb$ (e.g., $mb=1$):  With more frequent memory updates, the model focuses on short-range dependencies and produces more precise high-resolution spatial codes - i.e., better one-step prediction.
> >
> > (2) **Empirical evidence**. **Appendix Fig. A.2** (2D grid task) visualizes this trade-off. With small $mb$ (e.g., 1), mm-TEM generalizes well for short-range (single-step) prediction (**Fig. A.2A**) but performs worse on long-horizon prediction compared with mb = 4 or 8 (**Fig. A.2B**). With **large mb (e.g., 4 or 8)**, long-range prediction improves, while short-range accuracy decreases.
> >
> > This explains why **mm-TEM with $mb = 8$ performs well in multi-step imagination but not in single-step prediction**.
> >
> > (3) **Why Hippoformer does not suffer from this drawback**.  Hippoformer combines: a **Transformer** with a limited context window (good at short-step prediction), and an **mm-TEM module with mb = 8** (good at long-horizon prediction). Thus, the Transformer component compensates for the short-range weakness of mm-TEM.  In addition, employing a large mb in mm-TEM allows the memory MLP to leverage **highly parallelizable and efficient training** [5], and Hippoformer benefits from this increased training efficiency as well.
> >
> > 2. **Minor reference correction:** Thank you for pointing this out. Whittington et al. is indeed an ICLR 2022 paper "Relating transformers to models and neural representations of the hippocampal formation". We mistakenly cited it incorrectly here.
> >
> > [5] Ali Behrouz et al., NeurIPS, 2025

---

> > > ### Comment · Reviewer_Be6v · 2025-11-26
> > >
> > > Thank you for the explanation, it would be great to include add 1~2lines explaining the role of mm-TEM only and hippoformer  comparison in the main text.
> > >
> > > I raise my score form 6 to 8, good luck.

---

> > > > ### Author Response · Authors · 2025-11-26
> > > >
> > > > Thank you for your constructive suggestion and positive evaluation. We have now added brief lines clarification in the main text as requested.
> > > >
> > > > >Notably, for consistency with the Hippoformer, the standalone mm-TEM is also trained with $mb = 8$ on the 3D task. A large $mb$ inherently biases mm-TEM toward long-horizon prediction—enhancing multi-step imagination but reducing single-step accuracy (Tab. 1), a limitation that is compensated by the Transformer component in Hippoformer.
> > > >
> > > > We appreciate your time and are grateful for the score increase.

---

### Meta-Review · Area_Chair_K2Wb · 2026-01-09

**Summary:**

The paper proposes two models, mm-TEM and Hippoformer, inspired by hippocampus. Hippoformer is a hybrid of transformer and mm-TEM. The models are evaluated on observation prediction in “toy” (but still non-trivial) 2D environments and synthetic 3D navigation tasks. mm-TEM performs well in long-horizon prediction and learns grid-like representations of space. Hippoformer, by including transformer components, also does well on short-term prediction. On multi-step prediction it outperforms transformer and Titans baselines by a large margin.

Based on the reviews, the authors’ rebuttal, and the paper itself, the main strengths and weaknesses are as follows.

Pros:
1. Good motivation and presentation of the problem and the model
2. The model performs well in the 2D grid environments
3. In 3D prediction, on 1-step prediction, there’s no large difference, but for m-step prediction, the gap relative to the baselines is large
4. It’s good to see that the model discovers grid-like features
5. The authors did a good job with the rebuttal - one reviewer raised the score from 6 to 8, another one from 2 to X (the authors claim to 6, but I'm not quite sure where the 6 comes from)

Cons:
1. Evaluation only on fairly toy applications: 2D grid-like environments and 3D visually simple labyrinths.

Overall, the paper is somewhat borderline. There’s a lot going for it, but the evaluation settings are quite simplistic. In the end, given that the paper proposes a fairly new architecture, I believe it is okay that the evaluation is initially not the most realistic, as long as in the simple settings the model clearly has merit, which it does. So at this point I recommend acceptance, but I very much encourage the authors to apply the model to more realistic data - e.g. from robotics, driving, etc

**Reviewer Concerns:**

Reviewer 3ncY's concerns were the most detailed, and they were addressed satisfactorily, as the reviewer agrees.

**Reviewer Scores:**

As mentioned above, one reviewer raised the score from 6 to 8, another one from 2 to X (the authors claim to 6, but I'm not quite sure where the 6 comes from). I wouldn't expect further improvements

---

### Decision · Program_Chairs · 2026-01-26

Accept (Poster)